

# Long-range transport patterns into the tropical northwest Pacific during the CAMP²Ex aircraft campaign: chemical composition, size distributions, and the impact of convection

Miguel Ricardo A. Hilario[1,+], Ewan Crosbie[2,3], Michael Shook[2], Jeffrey S. Reid[4], Maria Obiminda L. Cambaliza[1,5], James Bernard B. Simpas[1,5], Luke Ziemba[2], Joshua P. DiGangi[2], Glenn S. Diskin[2], Phu Nguyen[6], Joseph Turk[7], Edward Winstead[2,3], Claire E. Robinson[2,3], Jian Wang[8], Jiaoshi Zhang[8], Yang Wang[9], Subin Yoon[10], James Flynn[10], Sergio L. Alvarez[10], Ali Behrangi[11,12], Armin Sorooshian[11,13,*]

[1] Manila Observatory, Quezon City 1108, Philippines
[2] NASA Langley Research Center, Hampton, VA, USA
[3] Science Systems and Applications, Inc., Hampton, VA, USA
[4] Marine Meteorology Division, Naval Research Laboratory, Monterey, CA, USA
[5] Department of Physics, Ateneo de Manila University, Quezon City 1108, Philippines
[6] Department of Civil & Environmental Engineering, University of California Irvine, Irvine, CA 92697, USA
[7] NASA Jet Propulsion Laboratory, Pasadena, CA, USA
[8] Center for Aerosol Science and Engineering, Department of Energy, Environmental and Chemical Engineering, Washington University in St. Louis, St. Louis, MO 63130, USA
[9] Department of Civil, Architectural and Environmental Engineering, Missouri University of Science and Technology, Rolla, MO 65409, USA
[10] Department of Earth and Atmospheric Science, University of Houston, Texas, 77204, USA
[11] Department of Hydrology and Atmospheric Sciences, University of Arizona, Tucson, AZ 85721, USA
[12] Department of Geosciences, University of Arizona, Tucson, AZ 85721, USA
[13] Department of Chemical and Environmental Engineering, University of Arizona, Tucson, AZ 85721, USA

*Correspondence to*: Armin Sorooshian (armin@email.arizona.edu)
+Now at: Department of Hydrology and Atmospheric Sciences, University of Arizona, Tucson, AZ 85721, USA





**Abstract**. The tropical Western North Pacific (TWNP) is a receptor for pollution sources throughout Asia and is highly susceptible to climate change, making it imperative to understand long-range transport in this complex aerosol-meteorological environment. Measurements from the NASA Cloud, Aerosol, and Monsoon Processes Philippines Experiment (CAMP²Ex; 24 Aug to 5 Oct 2019) and back trajectories from the National Oceanic and Atmospheric Administration Hybrid Single Particle Lagrangian Integrated Trajectory Model (HYSPLIT) were used to examine transport into the TWNP from the Maritime Continent (MC), Peninsular Southeast Asia (PSEA), East Asia (EA), and West Pacific (WP). A mid-campaign monsoon shift on 20 Sep 2019 led to distinct transport patterns between the southwest monsoon (before 20 Sep) and monsoon transition (after 20 Sep). During the southwest monsoon, long-range transport was a function of southwesterly winds and cyclones over the South China Sea. Low (high) altitude air generally came from MC (PSEA), implying distinct aerosol processing related to convection and perhaps wind shear. The monsoon transition saw transport from EA and WP, driven by Pacific northeasterly winds, continental anticyclones, and cyclones over the East China Sea. Composition of transported air differed by emission source and accumulated precipitation along trajectories (APT) as an indicator of convection. MC air was characterized by biomass burning tracers while major components of EA air pointed to Asian outflow and secondary formation. Convective scavenging of PSEA air was evidenced by considerable vertical differences between aerosol species but not trace gases, as well as notably higher APT and smaller particles than other regions. Finally, we observed a possible wet scavenging mechanism acting on MC air aloft that was not strictly linked to precipitation. These results are important for understanding the transport and processing of air masses with further implications for modeling aerosol lifecycles and guiding international policymaking on public health and climate.



## 1. Introduction

As pollution emissions from Asian countries have surpassed those of countries in Europe and North America (Akimoto, 2003; Smith et al., 2011), Asia becomes increasingly important from a global climate and health perspective. The tropical Western North Pacific (TWNP), situated adjacent to Southeast Asia (Fig. 1), is a receptor for multiple sources of aerosol particles throughout the region (Bagtasa et al., 2018; Hilario et al., 2020a; Huang et al., 2019; Reid et al., 2015) and is one of the most susceptible regions to global climate change (IPCC, 2014; Reid et al., 2013; Yusuf and Francisco, 2009). Amidst several multi-scale meteorological phenomena ranging from the Asian monsoon system (Akasaka et al., 2007; Chang et al., 2005), the El Niño Southern Oscillation (Cruz et al., 2013; Jose et al., 1996), the Madden-Julian Oscillation (Maloney & Hartmann, 2001; Pullen et al., 2015), and intermittent typhoons (Bagtasa, 2017; Maloney and Dickinson, 2003), the TWNP hosts arguably one of the most complex meteorological environments in the world with likewise intricate relationships to aerosol lifecycle and climate impacts (Reid et al., 2012; Ross et al., 2018).

Owing to atmospheric residence times ranging from days to weeks (Balkanski et al., 1993; Kritz and Rancher, 1980) and enabled by the surrounding meteorology, aerosol particles from multiple sources can undergo long-range transport into the TWNP (Lin et al., 2007; Xian et al., 2013). These sources include biomass burning from the Maritime Continent (MC) (Hilario et al., 2020a, 2020b; Reid et al., 2015), anthropogenic and dust outflow from East Asia (EA) (Bagtasa et al., 2019; Braun et al., 2020; Geng et al., 2019; Miyazaki, 2003; Oshima et al., 2012; Tan et al., 2012), emissions from Peninsular Southeast Asia (PSEA) (Bagtasa et al., 2019; Geng et al., 2019; Huang et al., 2020; Lin et al., 2009; Nguyen et al., 2020), and marine aerosol particles from the western Pacific (WP). Such transport is controlled by the interplay of several factors such as topography, sea breeze, monsoon flows, and typhoons (Reid et al., 2012; Wang et al., 2013b). Aside from the risk posed by transported anthropogenic aerosol on public health (Lelieveld et al., 2015; Zhang et al., 2017), such a diverse set of aerosol sources and types can result in variable aerosol-cloud-climate interactions (Hamid et al., 2001; Heald et al., 2014; Rosenfeld, 1999; Ross et al., 2018; Sorooshian et al., 2009; Yu et al., 2006; Yuan et al., 2011), which are complicated further by the spatial inhomogeneity of transported aerosol particles (Akimoto, 2003). As the influence of aerosol particles on climate remains one of the largest uncertainties in our understanding of the atmosphere (IPCC, 2014), investigating the composition and transport mechanisms of air masses from different source regions will aid in the future development of transport models and lead to a better understanding of the transport pathways that modulate aerosol particles in this part of the world.

Previous aircraft campaigns in Asia and the Pacific include the Transport and Chemical Evolution Over the Pacific (TRACE-P) campaign (Jacob et al., 2003), the Aerosol Radiative Forcing in East Asia (A-FORCE) campaign (Oshima et al., 2012), the Pacific Exploratory Mission – West A and B (PEM-West) (Hoell et al., 1996, 1997), and the Oxidant and Particulate Photochemical Processes Above a South East Asian Rainforest (OP3) project (Hewitt et al., 2010). These campaigns examined springtime outflow from the Asian continent (e.g., Koike et al., 2003; Kondo et al., 2004; Park, 2005) and early-summertime characteristics of local and transported aerosol over Borneo (e.g., Robinson et al., 2011, 2012); however, no study to our knowledge has utilized aircraft data to characterize long-range transport patterns over the TWNP coinciding with the peak agricultural burning period for Indonesia and Malaysia. Limited ship observations in association with the 7 Southeast Asian Studies (7SEAS) program (e.g., Reid et al., 2015, 2016a, 2016b) found a highly dynamic aerosol environment (Atwood et al., 2017; Hilario et al., 2020c; Reid et al., 2015).

The NASA Cloud, Aerosol, and Monsoon Processes-Philippines Experiment (CAMP²Ex) aircraft campaign examined the influence of meteorology, convection, and radiative effects on gas and aerosol species in the TNWP. Based at Clark, Luzon, Philippines, from 24 August to 5 October 2019, CAMP²Ex obtained a wide array of measurements between 0 – 9 km above ground level (AGL) across 19 research flights (RF). Some RFs were conducted in coordination with the seaborne research vessel R/V Sally Ride as part of the Office of Naval Research Propagation of InterSeasonal Tropical OscillatioNs (PISTON) project (https://onrpiston.colostate.edu/). The CAMP²Ex campaign is unique in that it began during the peak of the Asian southwest monsoon (SWM) and coincided with an early monsoon transition (MT), which occurred in late-September instead of the more common time in October (Chang et al., 2005; Matsumoto et al., 2020). The early MT led to diverse transport patterns (Fig. 2) that offered an opportunity to examine long-range transport into the TWNP.



Aircraft campaigns allow for vertically-resolved measurements of air mass properties, which are essential to better understand the atmosphere, as aerosol-cloud-climate interactions vary by altitude (Dahutia et al., 2019; Dong et al., 2017; Hansen, 2005; Mishra et al., 2015) and vertical transport can influence air mass composition (Matsui et al., 2011; Moteki et al., 2012; Oshima et al., 2012, 2013). Furthermore, as the main route of aerosol removal from the atmosphere, wet scavenging is a crucially important aspect of aerosol vertical profiles and are linked to significant uncertainties in climate models (Neu & Prather, 2012; Wang et al., 2013). Vertically-resolved in situ observations in field campaigns targeting aerosol-cloud-meteorology interactions are necessary to advance understanding of scavenging processes to inform the spectrum of models ranging from smaller-scale process models to larger-scale climate models (MacDonald et al., 2018; Sorooshian et al., 2019).

As Asian emissions continue to increase as a consequence of rapid economic development, it is imperative to understand the influence of long-range transport on air quality and aerosol-cloud-climate feedbacks in this region. In this study, we focus on characterizing transported air masses from four key regions: the Maritime Continent (MC: 5° S – 6.8° N, 94.9° E – 119.5° E), Peninsular Southeast Asia (PSEA: 10° N – 23° N, 95° E – 109.5° E), East Asia (EA: 22° N – 44° N, 100° E – 122° E and 30° N – 44° N, 122° E – 145° E), and the West Pacific Ocean (WP: 3° N – 25° N, 130° E – 145° E). Using air mass back trajectories to complement the CAMP²Ex data, this study aims to (1) identify regional transport pathways into the TWNP and their associated synoptic conditions, (2) characterize air masses coming from different source regions in terms of composition and aerosol size distribution, and (3) estimate the influence of convection and precipitation on transported air masses. By examining how transport and scavenging mechanisms impact air mass composition, our results have implications for improving modeling of aerosol lifecycles in this meteorologically complex region of the world and guiding policymaking related to public health and climate.

## 2. Methods

### 2.1. CAMP²Ex observations

A major goal of the 2019 CAMP²Ex aircraft campaign was to understand aerosol-cloud-climate feedbacks in the TWNP (Di Girolamo et al., 2018). Although multiple aircraft were deployed, this study focused on measurements made onboard the NASA P-3B Orion (N426NA) aircraft. Aircraft altitude (m AGL hereafter) was calculated as the difference between GPS altitude and ground elevation provided by the Google Maps API, with an uncertainty of ±5 m. Dry optical size distribution data were collected by the Laser Aerosol Spectrometer (LAS; TSI Model 3340) and are presented as an integrated particle number concentration for diameters between 100 and 1000 nm ($N_{100-1000nm}$; cm$^{-3}$). Uncertainty of $N_{100-1000nm}$ is estimated at 20%. Carbon monoxide (CO; ppm) and methane (CH$_4$; ppm) mixing ratios were measured by a dried-airstream near-infrared cavity ringdown absorption spectrometer (G2401-m; PICARRO, Inc.), with uncertainties of 2% and 1% and precisions of 0.005 ppm and 0.001 ppm, respectively. Ozone (O$_3$; ppbv) measurements were conducted with a dual beam UV absorption sensor (Model 205; 2B Technologies) with an uncertainty of 6 ppbv. Non-refractory aerosol composition in the submicrometer range was measured with a High-Resolution Time-of-Flight Aerosol Mass Spectrometer (AMS; Aerodyne) (DeCarlo et al., 2008). The species of relevance to this study include sulfate (SO$_4^{2-}$), ammonium (NH$_4^+$), nitrate (NO$_3^-$), and organic aerosol (OA), all of which are reported in units of µg m$^{-3}$ with uncertainties up to 50%. The AMS was operated in 1 Hz Fast-MS mode and averaged to 30-second time resolution for this study, with campaign-averaged 1-sigma detection limits (in ug m$^{-3}$) as follows: 0.169 (OA), 0.039 (SO$_4^{2-}$), 0.035 (NO$_3^-$), 0.169 (NH$_4^+$). Mass concentrations below these detection limit values, which are sometimes negative due to the AMS difference method, are statistically equal to zero. Black carbon (BC; ng m$^{-3}$) was measured with a Single-Particle Soot Photometer (SP2) (Moteki & Kondo, 2007, 2010), including an uncertainty of 10%. We note that BC data were unavailable during one flight covering a major Borneo smoke event (RF9); thus, the BC value presented in Table 1 is likely under-represented compared to the AMS data. A Fast Integrated Mobility Spectrometer (FIMS) measured aerosol size distribution between 10 nm and ~600 nm with a concentration uncertainty of ~ 15% and size uncertainty of ~ 3% (Wang et al., 2017a, 2017b, 2018a).

All aerosol data are reported at standard temperature and pressure (STP; 273 K, 1013 hPa). Only data collected from outside of clouds via isokinetic sampling (McNaughton et al., 2007) were used to eliminate sampling artifacts from the shattering of large water and ice particles (Murphy et al., 2004). Background concentrations of each species were defined as the lowest 10th percentile of all CAMP²Ex data along vertical profiles for every 5 K range of potential temperature (Koike et al., 2003; Matsui et al., 2011a). Enhancements above these background concentrations are



denoted by Δ. For species ratios, only data with ΔCO > 0.02 ppm were included similar to past work (Kleinman et al., 2007; Kondo et al., 2011; Matsui et al., 2011b).

Only data along profiles extending vertically more than 2 km were considered for analysis as they provide a "snapshot" of the atmosphere with which we can compare more directly air mass characteristics across different altitudes. Data collected when the P-3B aircraft sampled directly over urbanized Luzon (13° N – 15.8° N, 120° E – 122° E) were
excluded from analysis to minimize the impact of local emissions. Flight tracks and identified vertical profiles are shown in Fig. S1 of the Supplementary Information (SI).

*2.2. Back trajectory classification*
The National Oceanic and Atmospheric Administration (NOAA) Hybrid Single Particle Lagrangian Integrated Trajectory Model (HYSPLIT) (Rolph et al., 2017; Stein et al., 2015) was used to generate 120-hour back trajectories
along vertical profiles with one minute temporal resolution. Input meteorological data was from the National Centers for Environmental Prediction (NCEP) Global Forecast System (GFS) at a horizontal resolution of $0.25° \times 0.25°$.

Our analysis focused on transport from key source regions (MC, PSEA, EA, WP). We note here that "source region" refers to the attributed origin of an air mass as identified by our trajectory classification scheme and does not preclude the possibility of entrainment from emission sources during transport (e.g., shipping). Our classification scheme
considered two important environmental factors: (1) the synoptic shift that occurred around 20 Sep 2019, dividing the CAMP²Ex period into the SWM (before 20 Sep) and MT (after 20 Sep); and (2) the vertical wind shear across the region (Fig. 2). To better capture the pronounced effect of the monsoon shift, air masses were only classified as MC or PSEA (EA or WP) if sampled during the SWM (MT). For example, instances of EA air sampled during the SWM were classified as "Other" while air from EA sampled during MT were classified as EA. The inclusion of a monsoon
phase filter more explicitly highlights the temporal aspect of meteorology in the TWNP; however, without this monsoon phase criterion, resulting air mass classifications remain largely unchanged (Section 3.2). Furthermore, to account for regional vertical inhomogeneity (Atwood et al., 2013; Sarkar et al., 2018), our analysis of air mass characteristics differentiates between boundary layer (BL; < 2 km) and free troposphere (FT; > 2 km) (Section 3.3).

For an air mass to be classified into a source region, its back trajectory must pass within a source region's bounding
box for more than 6 h at an altitude below 2 km, which is the typical summertime BL height in the region (Chien et al., 2019). In addition to excluding data collected over urbanized Luzon (Section 2.1), trajectories passing through the Philippines (12° N – 18.25° N, 120.5° E – 122.5° E and 5.1° N – 14.5° N, 122.5° E – 126.7° E) under our trajectory classification scheme were excluded to further focus our analysis on long-range transport and associated processes.

Most air masses came from only one of the four source regions: WP (117 occurrences), MC (174 occurrences), PSEA
(88 occurrences), EA (130 occurrences). Air masses that passed through both EA and WP (12 occurrences) were considered as EA air due to the considerable influence of EA outflow on air mass composition (Talbot et al., 1997). Other transport permutations (e.g., air that passed through both MC and PSEA) did not meet the requirements of our classification scheme and were omitted. Thus, we focus on the four major transport pathways (MC, PSEA, WP, EA). Focusing on these major pathways adds robustness to the analysis by partly compensating for the limits of the
trajectory model in capturing more complex meteorological phenomena such as wind shear (Freitag et al., 2014), which have been shown to contribute to trajectory uncertainty (Siems et al., 2000; Stohl et al., 2002). In addition to requiring that the back trajectories pass through the source regions, the additional criteria imposed (e.g., altitude < 2 km over the source region) increase our confidence that the remaining cases represent instances of long-range transport. Resulting source region contributions per RF are shown in Fig. S2. We emphasize that these source region
contributions represent frequencies of observation rather than frequencies of occurrence, as the sampling location of the aircraft introduces a bias inherent in aircraft campaigns (Section 3.2).

As a consequence of our filtering scheme, a large portion of trajectories were tagged as "Other" (66.8%). This is attributable to several scenarios: (1) air masses that passed over source regions, but above our BL height threshold of 2 km; (2) air masses influenced by the Philippines (i.e., air masses that stayed over the Philippines at < 2 km for more
than 6 hours); (3) stagnant air masses that did not reach any source region; and (4) other transport permutations that occurred too infrequently to provide robust statistics.





Trajectory clustering was performed using two well-established methods: k-means and Ward linkage (Govender and Sivakumar, 2020) in order to confirm the robustness of our predefined source regions. K-means clustering classifies data into $k$ clusters such that the sum of squares per cluster is minimized (Hartigan and Wong, 1979), with the drawback that $k$ must be specified beforehand. Ward linkage is a hierarchical clustering method that merges clusters such that the increase in intra-cluster Ward's distance is minimized (Ward, 1963) and has been described as the method that most closely accomplishes the goals of clustering (Tufféry, 2011). More comprehensive descriptions of these clustering methods can be found elsewhere (Govender and Sivakumar, 2020; Pérez et al., 2017). Prior to clustering, a weighted distance matrix was calculated, similar to Taubman et al. (2006): (1) normalized trajectory coordinates to give equal weighting to both horizontal and vertical transport; (2) weighted time steps linearly back in time; and (3) assigned nearby points (time step < 6 h) zero weighting on the clustering to remove the influence of aircraft position on the clustering.

*2.3. Accumulated precipitation along individual trajectories*

Accumulated precipitation along individual trajectories (APT) was calculated using data from satellite precipitation products (SPPs): (1) the Precipitation Estimation from Remotely Sensed Information using Artificial Neural Networks – Climate Data Record (PERSIANN-CDR) (Ashouri et al., 2015; Nguyen et al., 2018); (2) the Integrated Multi-satellitE Retrievals for the Global Precipitation Measurement (GPM) mission (IMERG) (Huffman et al., 2019); and (3) the Tropical Rainfall Measuring Mission (TRMM) Multi-satellite Precipitation Analysis (TMPA) 3B42-V7 (Huffman et al., 2007). The purpose of utilizing multiple SPPs is to account for the uncertainties inherent in satellite retrievals, particularly during very light or heavy rainfall conditions, providing us with an ensemble of estimates rather than relying on a single SPP (Chen et al., 2020; Liu, 2016; Maggioni et al., 2016; Mahmud et al., 2017; Tan & Santo, 2018). Furthermore, although these SPPs measure surface precipitation and do not fully capture scavenging effects aloft, we use APT as an indicator of whether an air mass passed through a convectively active region.

PERSIANN-CDR ($0.25° \times 0.25°$, daily resolution) uses a modified PERSIANN algorithm utilizing NCEP Stage IV hourly precipitation and monthly precipitation from the Global Precipitation Climatology Project (GPCP) to maintain monthly amounts consistent with GPCP (Ashouri et al., 2015). PERSIANN-CDR data are available at the Center for Hydrometeorology and Remote Sensing (CHRS) Data Portal (http://chrsdata.eng.uci.edu) (Nguyen et al., 2019).

IMERG ($0.1° \times 0.1°$, 30-min resolution) integrates multiple satellite retrievals from the passive microwave (MW) precipitation retrievals provided by the suite of GPM constellation passive microwave radiometers (Kummerow et al., 2015), the Climate Prediction Center MORPHing technique (CMORPH) from NOAA, and PERSIANN-Cloud Classification System (PERSIANN-CCS; Hong et al., 2004) from the University of California, Irvine. These data are available from the NASA Precipitation Processing System (Skofronick-Jackson et al., 2018). For inter-calibrating various MW precipitation products, IMERG uses the GPM_2BCMB product (Olson et al., 2018) that utilizes the GPM Microwave Imager (GMI) and Dual-frequency Precipitation Radar (DPR) instruments on the GPM core satellite IMERG (Hou et al., 2014; Kidd et al., 2020). For this study, we use IMERG Final Run data, available at https://pmm.nasa.gov/data-access/downloads/gpm.

TMPA ($0.25° \times 0.25°$, 3-h resolution) similarly combines data from multiple satellites such as TRMM (pre-2015), NASA's *Aqua*, and the NOAA satellite series, involving calibration with gauge data when feasible (Huffman et al., 2007). Though TRMM ended its service in 2015, the TMPA 3B42 algorithm was continued in parallel with IMERG through December 2019 and had been based on a climatological calibration since 2014. As TMPA is climatologically calibrated, TMPA may be less sensitive to interannual variability in precipitation; thus, including TMPA in this study may provide a better idea of the spread among SPPs. TMPA data are accessible through https://pmm.nasa.gov/data-access/downloads/trmm.

Precipitation along each trajectory was integrated from the trajectory spawn point (i.e., P3-B sampling location) to the point when it reaches the boundary of a source region. An additional 24 h along the trajectory after reaching a source region was included in the APT integration to account for precipitation effects within the source region (Matsui et al., 2011a, b). No significant APT differences were found between using 0, 24, or 48 h for the APT calculation, suggesting that our results are robust with regard to the added duration. Furthermore, APT comparisons demonstrate that our results are independent of chosen SPP in terms of APT ranking (i.e., all SPPs agreed on which source regions are associated with the highest or lowest APTs).



*2.4. Other Data*

Elevation data (Fig. 1a) were acquired from the United States Geological Survey (USGS) Global Multi-resolution Terrain Elevation Data 2010 (GMTED2010) (Danielson and Gesch, 2011). Population density data for 2020 (Fig. 1b) were retrieved from the Gridded Population of the World (GPW), v4 (Center for International Earth Science Information Network, 2018) (https://sedac.ciesin.columbia.edu/data/set/gpw-v4-population-density-rev11/data-download). Depicted in Fig. 1c, active fires tagged with high confidence (> 80%; Bhardwaj et al., 2016) were obtained from the Moderate Resolution Imaging Spectroradiometer (MODIS) Collection 6 algorithm (https://firms.modaps.eosdis.nasa.gov/) (Levy et al., 2013) and converted into average fire density at $0.5° \times 0.5°$ resolution. Planetary BL (PBL) sulfur dioxide ($SO_2$) was retrieved by the Ozone Monitoring Instrument (OMI) and obtained from NASA Goddard Earth Sciences Data and Information Services Center (GES DISC) (Krotkov et al., 2015). The OMI $SO_2$ data were then resampled to $1° \times 1°$ resolution and averaged between 1 August and 15 October 2019 (Fig. 1d). Reanalysis data from NCEP/NCAR ($2.5° \times 2.5°$) were used to examine synoptic conditions (Kalnay et al., 1996).

**3. Results and Discussion**

*3.1. Observed transport patterns during CAMP²Ex*

Figure 1 provides an overview of the general source regions impacting the TWNP. The TWNP is surrounded by areas of high population density in EA, MC, and PSEA (Fig. 1b). Burning was prevalent mainly in the MC (Fig. 1c); though, fires were also detected along the eastern PSEA coast. Satellite retrievals of PBL $SO_2$ reveal possible point sources (Fig. 1d), perhaps from volcanoes, shipping, burning, and industry (Fioletov et al., 2016; Guttikunda et al., 2001; Zhang et al., 2019); however, we caution that cloud contamination may influence the $SO_2$ retrievals and are used here to demonstrate the variety of sources in Asia.

Trajectories from each source region show distinct pathways (Fig. 3; left column), indicative of differences in accompanying synoptic-scale circulation. These pathways are generally corroborated by both k-means and Ward linkage clustering methods (Figs. S3 and S4), confirming the robustness of our predefined source regions (Fig. 1a). Prior to further discussion, we emphasize the temporal aspect of these observed transport patterns (Figs. 2 and 3), in particular, their dependence on both synoptic (e.g., SWM) and mesoscale meteorology (e.g., typhoons), which varied during CAMP²Ex in terms of phase and frequency, respectively. Consequently, a specific transport pattern may be more dominant in one monsoon phase and less so in another while being enhanced (or suppressed) by intermittent mesoscale phenomena.

*3.1.1. Southwest monsoon*

Beginning with transport during the SWM prior to 20 Sep 2019, PSEA air is advected by uniform westerlies (Fig. 3a) associated with cyclonic activity over the northern South China Sea (SCS) (Fig. 3b) (Cheng et al., 2013; Huang et al., 2020; Lin et al., 2009). In comparison, although MC transport also occurs during the SWM, the mechanism behind MC transport is driven instead by southwesterlies originating across the MC (Fig. 3d) (Ge et al., 2014; Wang et al., 2013; Xian et al., 2013). Transport from the MC is further promoted by well-developed cyclones entering PSEA (Fig. 3e), as previously highlighted by observational (Hilario et al., 2020; Reid et al., 2015) and modeling studies (Wang et al., 2013). The similar cyclonic activity over northern SCS/PSEA may explain the confluence of air masses from PSEA and MC (e.g., RF6), indicated by the frequent sampling of MC and PSEA air in the same RF (Fig. S2).

A key difference between PSEA and MC air is that PSEA air passed through convective areas over the PSEA (Takahashi et al., 2010), the SCS (Fig. 3a, c) (Chen et al., 2017), and along the western coast of the Philippines (Akasaka et al., 2007; Chen et al., 2017; Cruz et al., 2013; Hilario et al., 2020b) while MC air passed through areas with less precipitation (Figs. 3d, f). As a result, PSEA air showed much higher APT than MC air (Table 1) and was more likely to have been processed by clouds. The transport pathway of PSEA through convective regions may lead to wet scavenging and aqueous-phase processing (MacDonald et al., 2018; Moteki et al., 2012; Sorooshian et al., 2006, 2007; Wonaschuetz et al., 2012), affecting both air mass composition and particle size distributions (Section 3.3).

In terms of sampled air masses, PSEA and MC showed contributions of 5.7% and 11.3%, respectively (Fig. 4a), and differ in terms of vertical distribution (Fig. 4b). PSEA air was sampled across a wide range of altitudes with the





majority of observations over 900 hPa, similar to Kondo et al. (2004), explainable by convection-related lofting (Fig.
5a). Very few observations of PSEA air were made near the surface. The lofting of PSEA air can occur over the PSEA
itself (Fig. 3c) (Kondo et al., 2004), through mechanisms like orographic effects (Lin et al., 2009). Convection over
the SCS trough likely also contributes to lofting (Fig. 3c) (Chen et al., 2017). Lofting of PSEA air into the FT has
important downstream ramifications as it modulates both aerosol composition and size distribution (Matsui et al.,
2011a; Moteki et al., 2012; Oshima et al., 2013). However, we note that vertical motion is an important source of
uncertainty in trajectory models (Harris et al., 2005) and should be interpreted with caution.

In contrast to PSEA air, sampled MC air was well-mixed within the BL (Fig. 5b), but the observation frequency of
MC air dropped sharply above 750 hPa, consistent with previous modeling studies (e.g., Xian et al., 2013). Such
distinct vertical distributions between MC and PSEA air are perhaps due to highly sheared environment during the
SWM, generally contributing to distinct air mass sources across different altitudes (Atwood et al., 2013; Sarkar et al.,
2018) and varying degrees to which these air masses are processed (Section 3.3).

*3.1.2. Monsoon transition*
The arrival of the MT period after 20 Sep 2019 led to a synoptic-scale shift (Fig. 2), allowing the sampling of air from
EA and WP (Fig. 3g, j). Transport from EA was observed across several MT flights (Fig. S2) and originated mainly
from southeastern China, Korea, and Japan (Fig. 3g), suggesting the entrainment of anthropogenic emissions (Section
3.3) (Cheng et al., 2013; Hatakeyama et al., 2001, 2004; Kim et al., 2009; Wang et al., 2016). Depicted in Fig. 3h,
Asian outflow was promoted by the pairing of a well-developed cyclone passing over the East China Sea (Hatakeyama
et al., 2001, 2004; Uno et al., 1998) and an anticyclone over the Asian continent (Honomichl and Pan, 2020). In
comparison, WP transport was observed mainly towards the end of the campaign (Fig. S2), likely a consequence of
sampling location, and was driven by Pacific northeasterlies (Figs. 3j – k). Transport from the WP, similar to that of
EA, coincided with an anticyclone over the Asian continent (Fig. 3k); however, WP transport is marked by the absence
of the East China Sea cyclone that promoted southward transport of EA air (Fig. 3h). This difference may explain why
EA and WP air were usually sampled in separate RFs (Fig. S2), in contrast to PSEA and MC air, which tended to be
sampled together.

Air from EA and WP show similarly low APT (Table 1), explainable by the generally lower precipitation in MT (Figs.
3i, l) compared to SWM (Figs. 3c, f) (Matsumoto et al., 2020), as well as the lower number of cyclone occurrences
after 20 Sep 2019. Although EA transport was driven by a well-developed cyclone (Fig. 3h), trajectories suggest that
EA air traveled through the outer bands of the cyclone (Fig. 3g), largely avoiding high precipitation areas (Fig. 3i).
This suggests that anthropogenic emissions entrained in EA air experienced low levels of scavenging and were more
likely to be sampled, in contrast with high APT urban source regions like PSEA (Section 3.3).

Transport from EA and WP were quite similar in terms of relative contribution (8.5% and 7.6%, respectively; Fig. 4a)
and vertical sampling distribution (Fig. 4b). Sampling of EA and WP air were largely constrained to the BL, though
sampled EA air was unimodal while WP air was more evenly sampled. In terms of vertical motion during transport,
some EA trajectories exhibited downward motion (Fig. 5c), likely due to subsidence from the continental anticyclone
(Fig. 3h), contrasting the vertical motion of PSEA air, which generally experienced upward motion associated with
convection (Fig. 5a).

In summary, important transport features over the TWNP include the following: (1) SWM transport from the MC and
PSEA was driven by southwesterlies and cyclonic activity over northern SCS/PSEA while MT transport from EA and
the WP was driven by Pacific northeasterlies, anticyclones over the Asian continent, and well-developed cyclones
over the East China Sea; (2) EA and MC air were sampled largely within the BL, did not exhibit significant upward
motion, and experienced low APT, suggesting that they likely carry urban/continental or biomass burning emissions;
in contrast, (3) PSEA air may have undergone a high degree of aerosol scavenging over convective regions (e.g.,
SCS), indicated by high APT and upward motion of trajectories.

*3.2. Sensitivity analysis*
In order to assess the uncertainty associated with our trajectory classification, we evaluated the effect of the following
variables on source classification: (1) trajectory height threshold over source regions; (2) back trajectory run time; (3)
vertical profile filtering; (4) monsoon phase; and (5) aircraft sampling location. Results are provided in Table S1 and





summarized below. Except for aircraft sampling location, independently changing any of these variables had little effect on the resulting source-region distribution. The relative contributions of source regions did vary significantly with sampling location, though areas surrounding Luzon (e.g., East of Luzon, North of Luzon) showed some degree of similarity. Thus, we emphasize that, as with any aircraft campaign, observed transport is to some degree dependent on aircraft location.

In order to reduce the effect of local emissions, we excluded trajectories classified as influenced by the Philippines (PH). To evaluate our filter for Philippine-influenced trajectories (hereafter, PH filter), air mass characteristics were compared between transported air unaffected by PH (No-PH air; e.g., MC) and transported air mixed with PH air (With PH; e.g., MC-PH). A local signal was observed for $N_{100-1000nm}$, suggested by differences in the histograms of $N_{100-1000nm}$ between non-PH and PH-mixed air (Fig. S5), particularly for MC and PSEA air. Differences in the species concentration histograms of non-PH and PH-mixed air were also observed for other anthropogenic species (BC, OA, $SO_4^{2-}$; not shown), confirming the effectiveness of the PH filter.

### 3.3. Chemical composition of transported air masses

A convenient opportunity afforded by having conducted the air mass classification presented above was to examine how gas and aerosol properties vary for each source region based on vertically-resolved in situ aircraft measurements. To account for regional vertical wind shear (Fig. 2) while considering the generally lower classification frequency at higher altitudes (Fig. 4b), air mass characterization was resolved into BL and FT subsets and composited by source region (Table 2). The delineation between BL and FT composition is demonstrated by selected species (Fig. 6), which generally dropped in concentration above the BL (< 850 hPa). Prior to further discussion, we note here that shipping is a major regional source (Streets et al., 1997, 2000) and may contribute appreciably to all air masses.

Significant differences in composition were observed in the same monsoon season (e.g., SWM) depending on air mass origin. Air from PSEA had much lower species concentrations than MC (Table 2) due to decreased emissions and increased potential for wet scavenging. Sampled MC air showed statistically significant differences between BL and FT concentrations for both gas and aerosol species (Table S2), indicative of emissions constrained to the BL, and exhibited strong biomass burning signals in its composition profile (e.g., $N_{100-1000nm}$, CO, $NO_3^-$, OA, and BC) (Maudlin et al., 2015; Pósfai et al., 2003; Reid et al., 1998, 2005; Theodoritsi et al., 2020; Yadav et al., 2017). We note that peaks of CO (Fig. 6b) and $CO_2$ (not shown) were observed in MC samples at around 650 hPa, suggestive of MC burning emissions lofted into the FT; however, this feature consisted of few samples and did not appear in other gases (e.g., $SO_2$) and thus warrants caution in its interpretation.

In contrast, PSEA air was generally characterized by concentration magnitudes between those of MC and WP. Aerosol concentrations of PSEA air in the FT were lower by at least an order of magnitude than those in the BL ($SO_4^{2-}$, $NH_4^+$, OA, BC) while trace gases (CO, $CH_4$) showed more similar BL and FT concentrations (Tables 2 and S2; Fig. S6). These aerosol-gas differences point to: (1) the lofting of PSEA air into the FT, as suggested by the similarity of trace gas concentrations between BL and FT, and (2) the consequent scavenging of aerosol particles, explaining the much lower aerosol concentrations in the FT (Oshima et al., 2012, 2013; Sievering et al., 1984). For comparison, MC air showed large BL and FT differences in both aerosol and gas species, the latter of which indicates the infrequent lofting of MC air (Figs. 4b and 5b). Since PSEA air came from a populated region (Fig. 1b) and likely originally contained anthropogenic aerosol particles, these unique characteristics of PSEA air compared to MC and EA air support the likelihood of aerosol scavenging in PSEA air. These observations are robust due to the relatively even sampling frequency of PSEA across altitudes (Fig. 4b).

Transport during the MT season showed similarly distinct composition profiles depending on air mass origin. Air from EA exhibited higher concentrations of $SO_4^{2-}$, $O_3$, $CH_4$, and $NH_4^+$, owing to urban emissions in continental outflow (Chuang et al., 2014; Talbot et al., 1997; Thornton et al., 1999; Umezawa et al., 2014; Wang et al., 2007) and extensive secondary aerosol formation (Hatakeyama et al., 2001, 2004, 2011; Krupa and Manning, 1988; Matsui et al., 2014). In contrast, WP air is characterized as pure marine due to composition similar to those previously reported in Pacific marine environments (Davis et al., 1996; Matsumoto et al., 1998; Talbot et al., 1997).





### 3.3.1. Species ratios

Composition profiles between regions (Table 2) reveal clear differences as a function of (1) emission source and (2) passage through convective regions indicated by APT (Table 1). The role of emission source was most evident when comparing air masses of low APT (EA, MC). Though EA and MC had similar BL values for $N_{100-1000nm}$ (Table 2), they showed distinct chemical differences: MC air was characterized by higher concentrations of biomass burning tracers (e.g., CO, $CH_4$, $NO_3^-$, OA) while EA air showed influence from urban/continental sources and secondary formation (e.g., $O_3$, $CH_4$, $NH_4^+$, $SO_4^{2-}$). Such differences in composition are corroborated by species ratios derived with linear regression (Fig. 7). Prior to a discussion on the species ratios, we note that the reported species ratios in this study are difficult to compare directly to at-source measurements of the same quantity because the composition of air masses was likely influenced by sources and sinks during transport (e.g., Choi et al., 2019; Conte et al., 2019; Gruber et al., 2019; Yang et al., 2009); however, differences in species ratios can still aid in air mass characterization and point to possible emission sources.

In Fig. 7a, the lower $\Delta CO/\Delta CO_2$ ratio of EA air versus MC air (Fig. 7a) is indicative of an inefficient combustion signature in MC air (Halliday et al., 2019), attributable to the predominantly smoldering phase of MC fires (Gras et al., 1999; Reid et al., 2013). We note that the poor $\Delta CO$-$\Delta CO_2$ correlation for MC air indicates that our reported ratio does not reflect expected emission ratios (Andreae, 2019; Hurst et al., 1994). This further suggests (1) our source region classification (i.e., MC, EA) may not perfectly capture air mass differences, and (2) additional sources of CO or $CO_2$ during transport. Thus, it is necessary to use multiple species ratios to supplement air mass chemical characterization.

The strong relationship between $CH_4$ and CO in MC air is a good indication of biomass burning influence (Andreae, 2019; Hecobian et al., 2011). The ratio of $\Delta CH_4/\Delta CO$ (Fig. 7b) was much higher in EA air compared to MC air, indicating the dominance of $CH_4$ from residential and industrial activity (Geng et al., 2019; He et al., 2019; Tohjima et al., 2014) as well as from rice cultivation in EA (Xia et al., 2020).

As an indicator of aerosol hygroscopicity (Kreidenweis and Asa-Awuku, 2014; Malm et al., 2005; Svenningsson et al., 2006), the $\Delta OA/\Delta SO_4^{2-}$ ratio pointed to higher hygroscopicity in EA air than MC air (Cheung et al., 2020; Saxena et al., 1995; Wang et al., 2017, 2018, 2019). Interestingly, though peat burning in the MC has been associated with $SO_4^{2-}$ (Ikegami et al., 2001; Reid et al., 2013), the elevated $\Delta OA/\Delta SO_4^{2-}$ ratio in MC air ($4.85 \pm 0.24$) is indicative of lower hygroscopicity, explainable by the high levels of OA emitted during biomass burning (Radzi bin Abas et al., 2004; Theodoritsi et al., 2020) or produced through gas-to-particle conversion during transport (Cappa et al., 2020; Mardi et al., 2018; Zhou et al., 2012). The lower $\Delta OA/\Delta SO_4^{2-}$ ratio of EA air ($0.75 \pm 0.09$) signaling high hygroscopicity is attributable to the high levels of $SO_2$ in EA (Fig. 1d), leading to the secondary formation of $SO_4^{2-}$ (e.g., Hatakeyama et al., 2011).

Although MC was calculated to have low APT (Table 1), a comparison of BL and FT air from MC (Figs. S6 and S7) allows for speculation on a possible scavenging mechanism acting on FT air. Linear regressions of $\Delta SO_4^{2-}/\Delta CO$ (Fig. S7a) suggested the removal of $SO_4^{2-}$ from FT air while $\Delta OA/\Delta CO$ (Fig. S7b) indicated no such effect on OA. Considering the higher hygroscopicity and therefore scavenging susceptibility of $SO_4^{2-}$ compared to OA (e.g., Kreidenweis and Asa-Awuku, 2014), we speculate the removal of $SO_4^{2-}$ to be related to wet scavenging. Comparisons of BL and FT species concentrations (Fig. S6) further supports the possibility of this hypothesized scavenging, as aerosol species have significantly lower concentrations in the FT compared to BL, a difference not observed for trace gases. The potential for scavenging was also supported by number size distributions (Section 3.4).

We note that this wet scavenging mechanism is not apparent from APT, which suggested dry conditions for MC air (Table 1). This disagreement with APT stems from the usage of SPPs which typically describe surface precipitation and, consequently, our APT may not fully capture potential wet scavenging effects aloft. Thus, the speculated scavenging mechanism of MC air in the FT may occur at higher levels and may not be strictly linked to surface precipitation. Perhaps, this mechanism is related to processes such as in-cloud scavenging (Sievering et al., 1984; Yang et al., 2020; Yu et al., 2020) but, indeed, this requires a deeper investigation in future work.

Due to BC's lack of secondary sources, the ratio of $\Delta BC/\Delta CO$ has been used to gauge transport efficiency as affected by physical removal processes on air masses (Matsui et al., 2011b; Moteki et al., 2012; Oshima et al., 2012) and as an



indicator of combustion efficiency, which increases with ΔBC/ΔCO (Kondo et al., 2011). The ratio of ΔBC/ΔCO was much higher in EA air than in MC air (Fig. 7f), similar to observations by Pani et al. (2019) in Taiwan. This difference is explainable by burning in industrial and residential areas in EA (Bond et al., 2004; Geng et al., 2019) and the predominance of smoldering fires in the MC (Gras et al., 1999; Reid et al., 2013), which yield a lower ΔBC/ΔCO than flaming fires (Kondo et al., 2011).

*3.4. Size distributions of transported air masses*
To more deeply characterize the air masses from different source regions, we examine the differences in normalized FIMS number (Fig. 8) and volume (Fig. S8) size distributions between BL and FT, which can also offer insights into the convection-related removal of PSEA air. In-cloud processing during transport may influence particle sizes in these air masses, whereby a combination of the following processes can occur (e.g., Ervens, 2015; Sorooshian et al., 2007;
Wonaschuetz et al., 2012), followed by detrainment from the cloud or wet removal: (1) collisions between interstitial aerosol and droplets; (2) coalescence among droplets; and (3) aqueous-phase processing in droplets. However, comparisons between size distributions between regions and between BL and FT may still lend valuable insights into transport-related processes (e.g., Moteki et al., 2012).

Firstly, comparing source regions of low APT and high aerosol concentration, EA air in the BL (Fig. 8a) showed a
wider peak in its size distribution (40 – 200 nm) than that of MC (Fig. 8b), which showed a clear unimodal peak (100 nm). This was perhaps an effect of multiple sources contributing to EA air masses, ranging from industrial activities to rice cultivation (Chen et al., 2020b; Geng et al., 2019; Wang et al., 2016; Xia et al., 2020). In comparison, biomass burning emissions from the MC have been shown to greatly influence air mass composition (Engling et al., 2014; Fujii et al., 2015; Santoso et al., 2011) and, by extension, such a dominant emission source can explain the large
unimodal peak in MC's BL size distribution (Figs. 8b and S8b).

A comparison of the FT and BL size distributions of MC air may point to a potential scavenging mechanism of MC air lofted into the FT, signaled by significant BL and FT differences above 50 nm (Fig. 8b). This potential scavenging mechanism of MC air was previously proposed using species ratios (Section 3.3.1), and the size distribution here provides further evidence for the hypothesized removal process; however, we emphasize that the hypothesized
scavenging mechanism is for now speculative and warrants future examination.

In contrast, the size distribution of PSEA air in the BL shows two peaks at around 50 nm and 200 nm. Comparing PSEA and MC air in the BL reveals much smaller particle sizes in PSEA air (Fig. 8c), explainable by differences in source emissions as well as in APT. A comparison of BL and FT air from PSEA pointed to scavenging during lofting into the FT: the FT size distribution of PSEA air showed a sharp drop in particle number concentrations above 50 nm
while the BL size distribution of PSEA air was much broader. In fact, the size distribution of FT air from the PSEA is more similar in shape to that of WP air (Fig. 8d), which is representative of background FT air. Due to this similarity, the peaks at 30 nm in FT air from PSEA and WP may originate from new particle formation (Williamson et al., 2019) that has been shown to be connected to APT in marine environments (Ueda et al., 2016), such as the convectively active SCS.

*3.5. Influence of convection on transported air masses*
The relationship between composition and convection was further investigated through scatterplots of ΔBC/ΔCO ratio, an indicator of physical removal processes (Moteki et al., 2012), as a function of APT, an indicator of convection. The decrease in ΔBC/ΔCO ratio with APT (Fig. 9a) indicates that convection during transport is one of the main scavenging mechanisms in the TWNP. Both EA and MC air showed very high ΔBC/ΔCO ratios compared to PSEA,
indicative of more efficient transport, while having low APT, which allowed for a clear transport signal as shown by the high concentrations of anthropogenic or burning species in these air masses (Section 3.3). In contrast, PSEA air is characterized by a lower ΔBC/ΔCO ratio coinciding with high APT (Fig. 3c; Table 1), pointing to particle scavenging over convective areas. To further demonstrate the impact of convection on transport efficiency, Fig. 9b reveals ΔBC/ΔCO distributions resolved by low and high APT. The shift towards higher ΔBC/ΔCO ratios under low APT
implies that dry and non-convective conditions are conducive to transport, suggesting the higher BL concentrations of anthropogenic species in MC, EA air were likely enabled by lower levels of wet removal.





*4. Summary and Conclusions*

Utilizing airborne CAMP²Ex measurements between 24 Aug and 5 Oct 2019 and HYSPLIT back trajectories, we examined transport patterns into TWNP from key source regions (PSEA, MC, EA, WP). Key conclusions from this study include the following:

1. Meteorological phenomena driving transport as well as the origins of transported air masses shifted significantly with the monsoon phase. During the SWM, MC and PSEA transport were associated with monsoon-driven southwesterly winds and cyclonic activity over the northern SCS. Wind shear was associated with predominantly BL (FT) sampling of MC (PSEA) air, implying distinct aerosol processing between these two source regions. In comparison, transport during the MT period from EA and WP was driven by northeasterly winds from the Pacific, anticyclones over the Asian continent, and well-developed cyclones passing through the East China Sea. These transport differences led to varying degrees of convection experienced by transported air masses. PSEA air generally passed through convective regions (SCS, west of Luzon, and over the PSEA itself) and was lofted into the FT, which led to scavenging of aerosols. In contrast, air masses from the MC and EA underwent relatively little convection, indicated by low APT, and mainly were confined to the BL, enabling the transport of anthropogenic emissions.

2. Characteristics of transported air masses differed primarily by emission source and passage through convective regions. Due to low APT and high $\Delta BC/\Delta CO$, transported air from MC and EA exhibited characteristic emissions: MC air from biomass burning (CO, well-correlated CO and $CH_4$, $NO_3^-$, OA) and EA air from anthropogenic outflow and secondary formation ($O_3$, $CH_4$, $NH_4^+$, $SO_4^{2-}$). Key species ratios corroborated distinct sources between MC and EA air. Aerosol size distributions in EA air suggested multiple primary sources (industry, residential emissions, rice cultivation) as well as secondary formation, indicated by its relatively broad peak; in contrast, the narrower peak in the size distribution of MC air pointed to the predominance of biomass burning emissions.

3. Air from the PSEA showed strong evidence of particle scavenging: passage over high precipitation areas, convective lofting, high APT, low $\Delta BC/\Delta CO$, relatively low levels of anthropogenic species, and a size distribution shifted towards smaller particle sizes. Aerosol concentrations of PSEA air in the FT were lower by at least an order of magnitude than those in the BL, a difference that was not observed for trace gases, which pointed to scavenging of aerosol particles in the FT. Furthermore, PSEA air in the FT lacked the larger peak ($D_p$ = 200 nm) observed in BL air and instead peaked at much smaller sizes ($D_p$ = 30 nm), suggesting large particle removal during convective lofting. The fine mode peak ($D_p$ = 30 nm) for PSEA FT air also resembled that of WP air, suggestive of new particle formation during transport from the PSEA, perhaps occurring over the convective SCS.

4. A possible wet scavenging mechanism for MC FT air was inferred from $\Delta SO_4^{2-}/\Delta CO$ and $\Delta OA/\Delta CO$ ratios between BL and FT, and corroborated by size distributions, which showed significant BL and FT differences for larger particles (> 50 nm). The disagreement with APT was attributed to SPP limitations in capturing scavenging effects aloft, hinting that the scavenging mechanism acts at higher layers and may not be linked to surface precipitation. However, we emphasize that the exact scavenging mechanism is for now speculative and warrants its own investigation in the future.

Recommendations for future work include: (1) investigating the hypothesized scavenging mechanism of MC air aloft using vertically-resolved moisture and convection retrievals; (2) examining deep convection periods to further evaluate wet scavenging effects on transported air; (3) characterizing aerosol hygroscopicity as a function of air mass source region and transport processes; and (4) comparing different sampling areas over the Philippines as impacted by transported air masses.

*Data Availability.* The CAMP²Ex dataset can be found at https://doi.org/10.5067/Suborbital/CAMP2EX2018/DATA001 (last access: 11 September 2020). HYSPLIT data are accessible through the NOAA READY website (http://www.ready.noaa.gov; last accessed: 13 July 2020). Global elevation data from GMTED2010 are available at http://temis.nl/data/gmted2010/ (last access: 12 March 2020). Population density data are provided by the Center for International Earth Science Information Network, available at https://sedac.ciesin.columbia.edu/data/set/gpw-v4-population-density-rev11/data-download; last access: 3 July 2020). MODIS active fire data can be downloaded through the Fire Information for Resource Management System



(https://firms.modaps.eosdis.nasa.gov/; last access: 29 June 2020). OMI data were retrieved from the NASA GES
DISC website (https://doi.org/10.5067/Aura/OMI/DATA3008; last access: 27 June 2020). NCEP/NCAR Reanalysis
data were provided by the NOAA/OAR/ESRL PSD, Boulder, Colorado, USA, from their website
(https://www.esrl.noaa.gov/psd/; last access: 13 June 2020).

*Author contributions.* MRAH performed the analysis and prepared the manuscript. All authors provided input for the
manuscript and/or participated in data collection and processing.

*Competing interests*. The authors declare that they have no conflict of interest.

*Acknowledgements*. This research was funded by NASA grant 80NSSC18K0148 in support of CAMP²Ex.

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





**Table 1: Statistics for accumulated precipitation along trajectories (APT, mm) for Peninsular Southeast Asia (PSEA), Maritime Continent (MC), East Asia (EA), and West Pacific (WP), calculated with IMERG, TMPA, and PERSIANN-CDR. Median values are presented along with the 25th and 75th percentiles provided in parentheses.**

|  | IMERG | TMPA | PERSIANN-CDR |
|---|---|---|---|
| **PSEA** | 34.11 (19.47 – 49.13) | 39.83 (22.25 – 60.08) | 34.74 (28.89 – 43.84) |
| **MC** | 1.70 (0.34 – 11.78) | 2.15 (0.00 – 12.64) | 6.10 (3.11 – 17.30) |
| **EA** | 0.54 (0.04 – 1.48) | 1.23 (0.14 – 2.94) | 5.14 (2.67 – 12.13) |
| **WP** | 1.31 (0.12 – 7.45) | 3.17 (0.00 – 13.17) | 14.30 (4.78 – 20.05) |



**Table 2: Comparisons of boundary layer (BL; < 2 km) and free troposphere (FT; > 2 km) mean values across source regions. Uncertainty values are presented as 1**
**standard deviation. Bold values denote when significant statistical differences are found ($p < 0.05$). Corresponding p-values are provided in Table S2 of the Supplementary**
**Information. The EA-FT column was left blank due to the infrequent sampling of EA air in the FT.**

| | EA | | MC | | PSEA | | WP | |
|---|---|---|---|---|---|---|---|---|
| | **BL** | **FT** | **BL** | **FT** | **BL** | **FT** | **BL** | **FT** |
| $N_{100-1000nm}$ (cm$^{-3}$) | 839.11 ± 507.34 | - | **818.43 ± 571.90** | **223.87 ± 316.26** | **272.55 ± 125.29** | **35.88 ± 41.23** | **71.18 ± 66.13** | **14.72 ± 5.99** |
| **CO (ppm)** | 0.16 ± 0.07 | - | **0.18 ± 0.12** | **0.11 ± 0.07** | 0.10 ± 0.02 | 0.10 ± 0.02 | 0.08 ± 0.00 | 0.08 ± 0.01 |
| **O$_3$ (ppbv)** | 45.29 ± 28.89 | - | **31.22 ± 10.62** | **23.69 ± 7.62** | **24.03 ± 3.44** | **29.69 ± 5.52** | **12.73 ± 3.27** | **18.81 ± 7.79** |
| **CH$_4$ (ppm)** | 1.96 ± 0.06 | - | **1.86 ± 0.01** | **1.85 ± 0.01** | 1.88 ± 0.03 | 1.87 ± 0.01 | 1.88 ± 0.00 | 1.88 ± 0.01 |
| **SO$_4^{2-}$ (µg m$^{-3}$)** | 5.14 ± 2.40 | - | **2.43 ± 1.33** | **0.69 ± 0.75** | **1.03 ± 0.54** | **0.17 ± 0.15** | **0.79 ± 0.98** | **0.23 ± 0.09** |
| **NO$_3^-$ (µg m$^{-3}$)** | 0.19 ± 0.45 | - | **0.24 ± 0.32** | **0.08 ± 0.20** | 0.04 ± 0.04 | 0.00 ± 0.03 | 0.01 ± 0.04 | -0.01 ± 0.05 |
| **NH$_4^+$ (µg m$^{-3}$)** | 1.32 ± 1.19 | - | **0.86 ± 0.75** | **0.28 ± 0.43** | 0.23 ± 0.24 | -0.01 ± 0.18 | 0.10 ± 0.30 | 0.00 ± 0.20 |
| **OA (µg m$^{-3}$)** | 2.68 ± 3.54 | - | **7.23 ± 8.80** | **2.10 ± 4.52** | **0.96 ± 1.00** | **0.07 ± 0.24** | **0.13 ± 0.32** | **0.01 ± 0.21** |
| **BC (ng m$^{-3}$)** | 87.29 ± 98.53 | - | **71.81 ± 41.79** | **16.06 ± 17.40** | **24.90 ± 15.83** | **2.79 ± 5.59** | **1.03 ± 2.92** | **0.20 ± 0.91** |



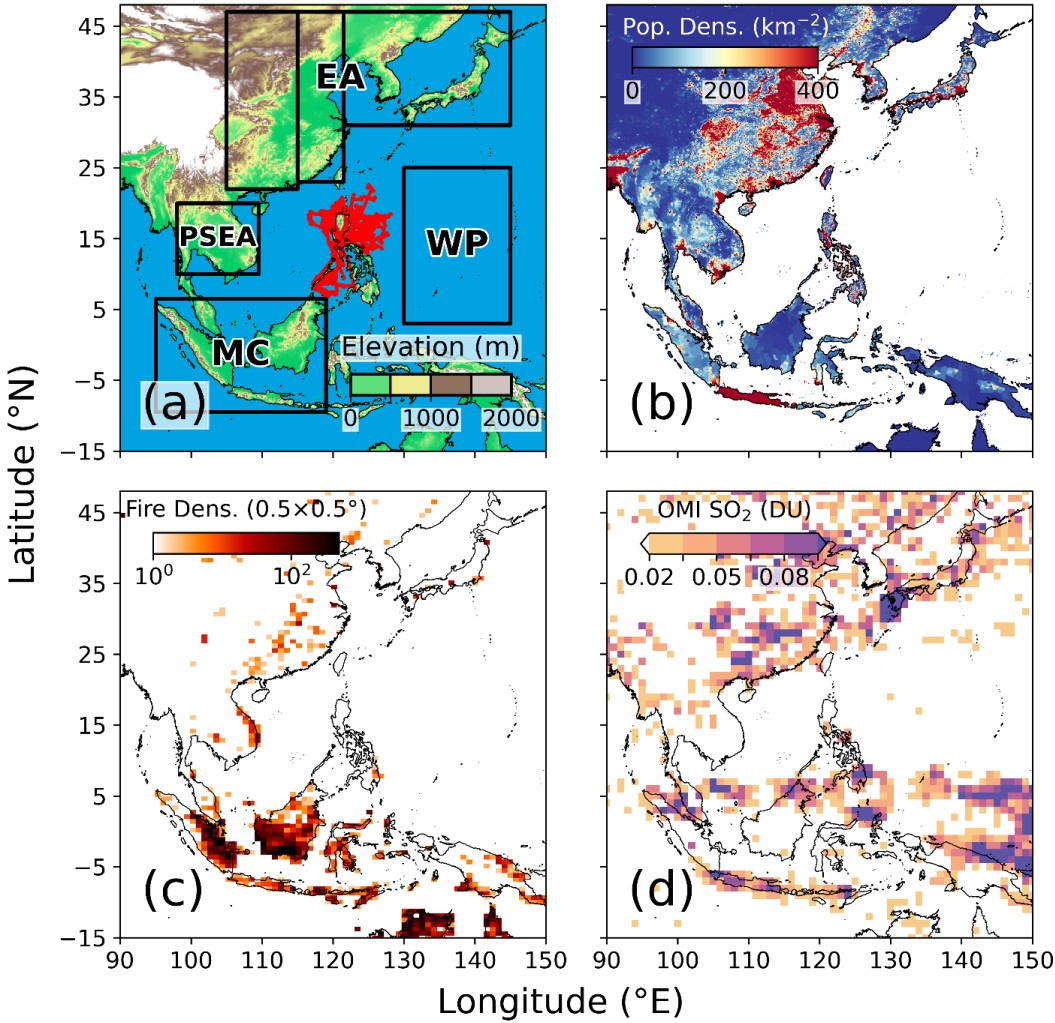

**Figure 1: Maps of (a) ground elevation from the Global Multi-resolution Terrain Elevation Data 2010 (GMTED2010), flight**
**tracks in red, and approximate source regions in labeled boxes: Peninsular Southeast Asia (PSEA), Maritime Continent**
**(MC), East Asia (EA), and West Pacific (WP), (b) 2020 population density from the Center for International Earth Science**
**Information Network (CIESIN) Gridded Population of the World (GPW) v4, (c) MODIS active fire hotspot density (only**
**fires tagged with > 80% confidence) averaged at 0.5° × 0.5° resolution from 1 Aug to 15 Oct 2019, and (d) OMI-retrieved**
**PBL SO$_2$ averaged over the same period.**



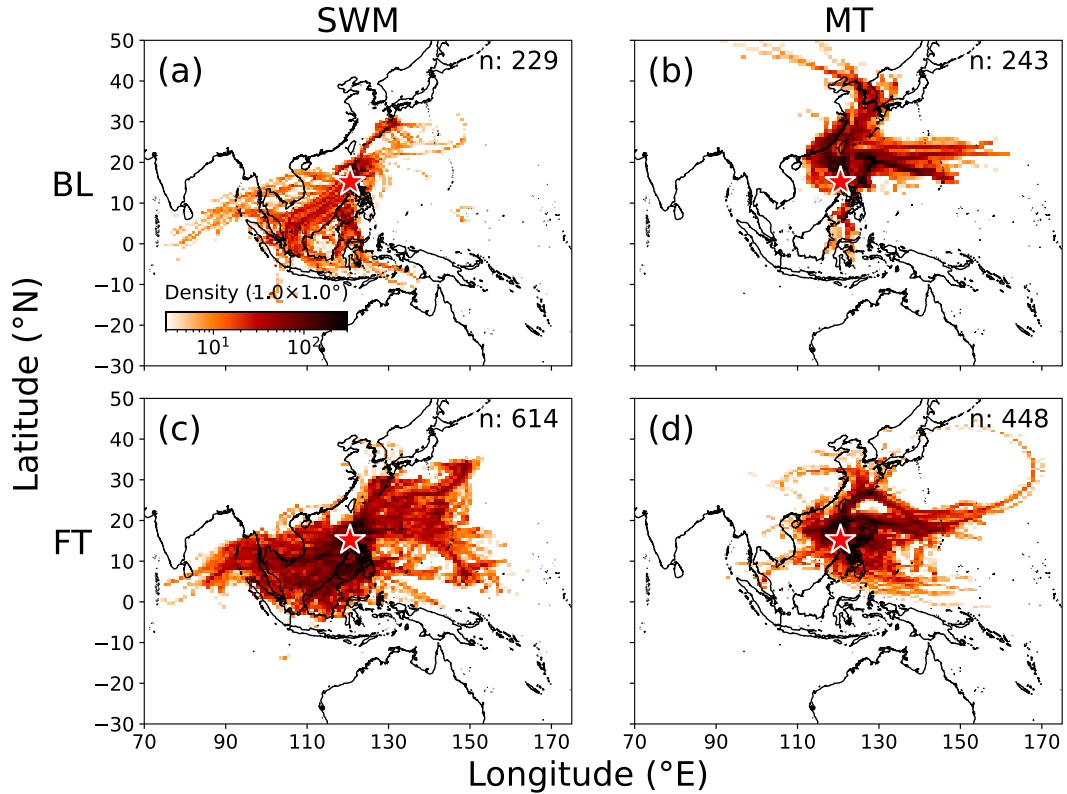

**Figure 2: Trajectory densities resolved by monsoon phase and sampling vertical level. Monsoon phases are southwest monsoon (SWM; before 20 Sep 2019) and the monsoon transition (MT; after 20 Sep 2019). Vertical sampling levels are divided into boundary layer (BL; < 2 km) and free troposphere (FT; > 2 km). Red star denotes Clark International Airport, Philippines. Number of trajectories (n) is shown in the upper right of each panel.**









**Figure 3: Classified trajectories and synoptic conditions associated with transport from (a – c) Peninsular Southeast Asia
(PSEA), (d – f) Maritime Continent (MC), (g – i) East Asia (EA), and (j – l) West Pacific (WP). Left: Trajectory density
normalized to the mean per source region, with the number of trajectories classified into each source region annotated on
the lower left of each panel. Middle: NCEP 850 mb geopotential height anomaly from the mean for 2019, overlaid with
horizontal winds (≥ 2 m s⁻¹). Right: PERSIANN-CDR average precipitation overlaid with NCEP 850 mb ω where red (blue)
contour lines represent ω values above 0.05 Pa s⁻¹ (below -0.05 Pa s⁻¹).**

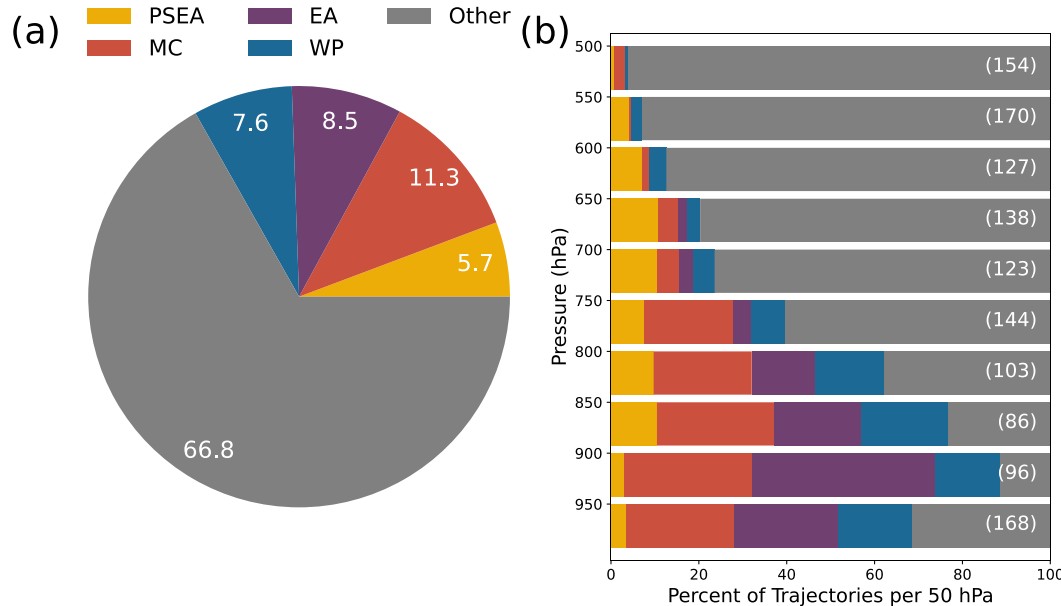

**Figure 4: Relative contributions (in percentages) of air masses arriving from study regions (a) averaged over all altitudes
and (b) pressure levels (hPa). Total number of trajectories per pressure bin is provided on the right end of (b). Source
regions are Peninsular Southeast Asia (PSEA), Maritime Continent (MC), East Asia (EA), and West Pacific (WP).**



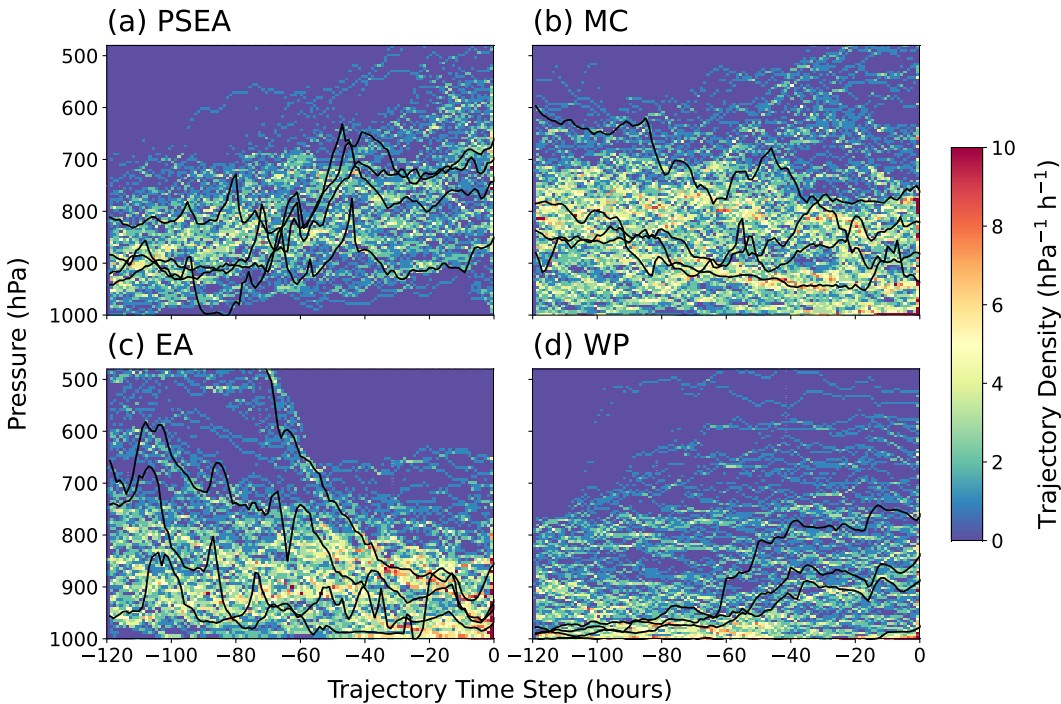

**Figure 5: Vertical motion during transport from (a) Peninsular Southeast Asia (PSEA), (b) Maritime Continent (MC), (c)**
**East Asia (EA), and (d) West Pacific (WP). Color corresponds to density as a function of trajectory altitude (pressure) and**
**time step. Example trajectories are plotted in black to show actual vertical motion.**





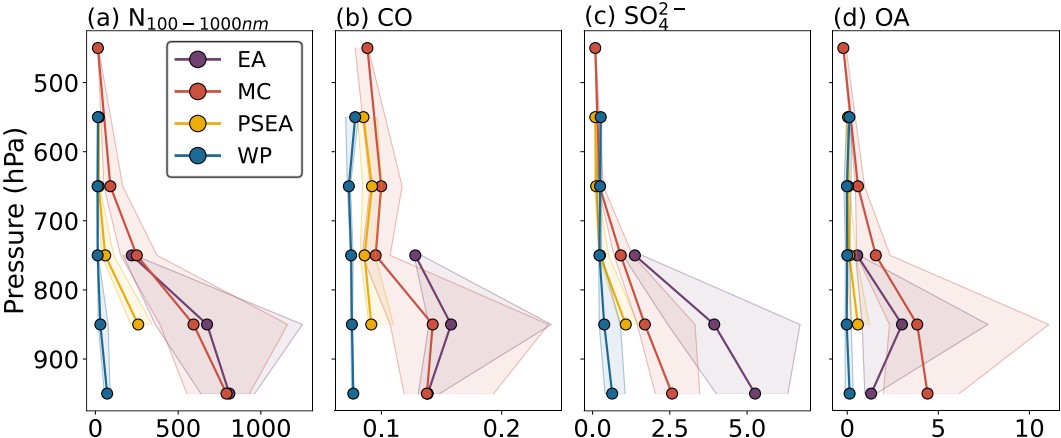

**Figure 6: Vertical median profiles of composition for Peninsular Southeast Asia (PSEA), Maritime Continent (MC), East**
**Asia (EA), and West Pacific (WP). (a) $N_{100-1000nm}$ (cm$^{-3}$), (b) CO (ppm), (c) $SO_4^{2-}$ (μg m$^{-3}$), (d) OA (μg m$^{-3}$). Left and right**
**sides of shaded areas refer to 25th and 75th percentiles, respectively.**



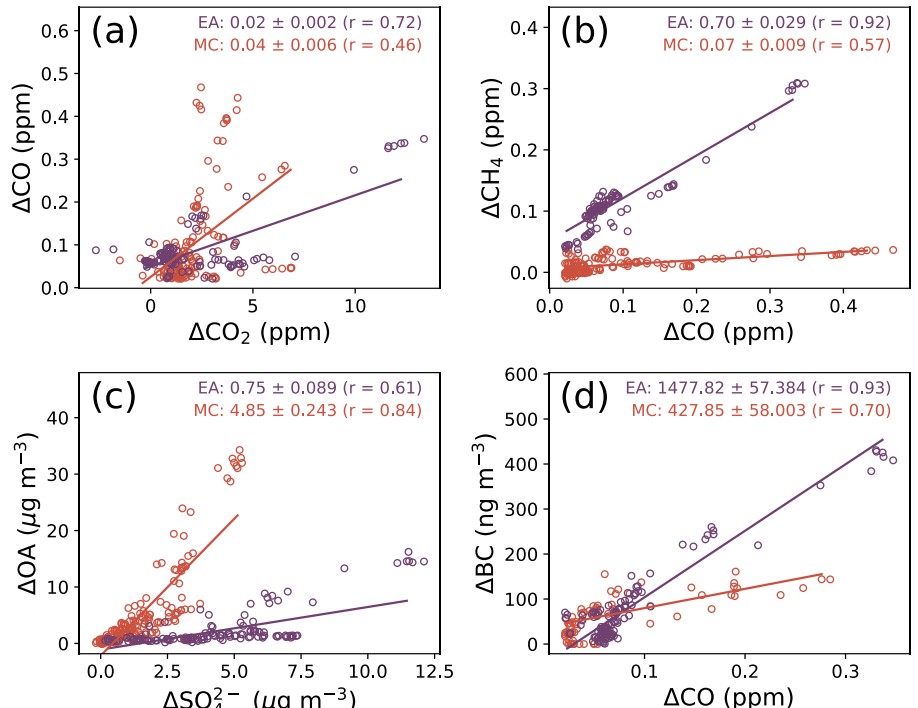

1249

**Figure 7: Linear regressions of (a) ΔCO/ΔCO₂, (b) ΔCH₄/ΔCO₂, (c) ΔOA/ΔSO₄²⁻, and (d) ΔBC/ΔCO for the Maritime Continent (MC) and East Asia (EA). Annotated are species ratios (slope) per region and standard error (SE) as a measure of uncertainty (slope ± SE). Pearson's R values are provided in parentheses. Only data with ΔCO > 0.02 ppm were included to better identify combustion-related ratios. Peninsular Southeast Asia and West Pacific data were not plotted due to low correlations.**



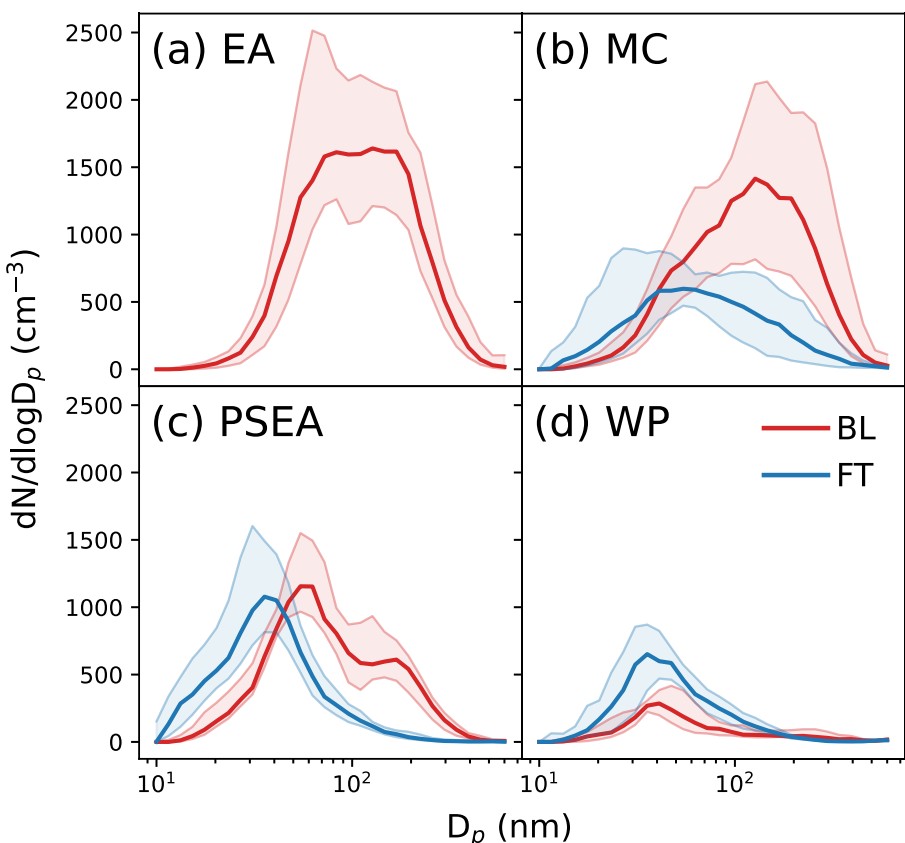

**Figure 8: Median FIMS number size distributions (dN/dlogD$_p$; cm$^{-3}$) as a function of geometric mean particle diameter (D$_p$; nm), resolved by source region and sampling altitude in the boundary layer (BL; < 2 km) and free troposphere (FT; > 2 km). Source regions are Peninsular Southeast Asia (PSEA), Maritime Continent (MC), East Asia (EA), and West Pacific (WP). Upper and lower bounds of the shaded areas refer to 25$^{th}$ and 75$^{th}$ percentiles, respectively. The size distribution of EA air in the FT was not plotted due to the infrequent sampling of EA air in the FT.**

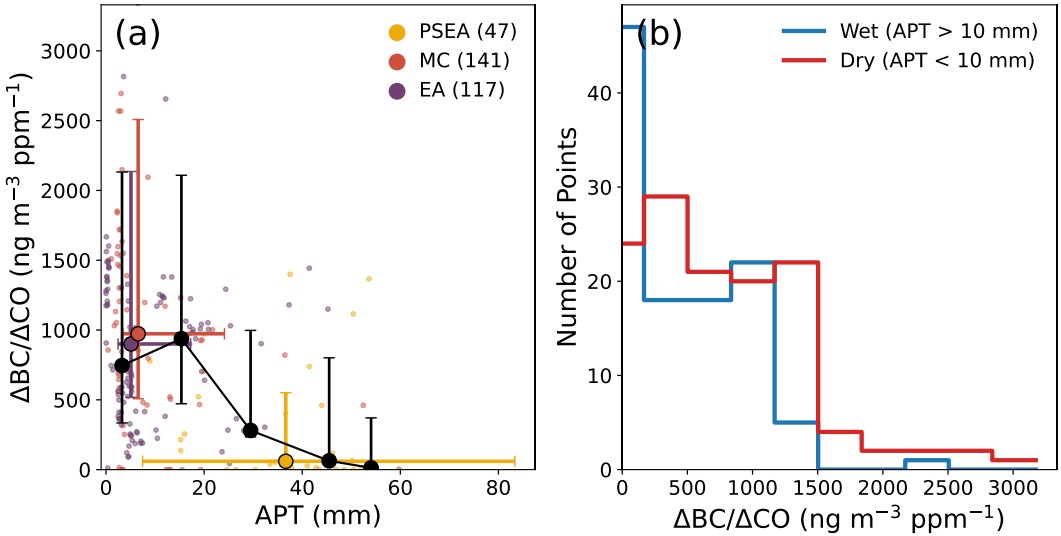

**Figure 9: (a) ΔBC/ΔCO ratio (ΔCO > 0.02 ppm) as a function of accumulated precipitation along individual trajectories (APT) calculated with PERSIANN-CDR, and (b) histograms of ΔBC/ΔCO resolved by wet (APT > 10 mm) and dry (APT < 10 mm) air masses. Medians and 25th/75th percentiles (error bars) are shown for (a) each source region: Peninsular Southeast Asia (PSEA), Maritime Continent (MC), and East Asia (EA). West Pacific (WP) was not plotted because of few data points where ΔCO > 0.02 ppm. Black line in (a) represents the trend for all source-classified data (EA, MC, PSEA, WP) grouped into APT bins. Number of observations plotted per source region are provided in parentheses in (a).**