# Peer review of "Long-range transport patterns into the tropical northwest Pacific during the CAMP2Ex aircraft campaign: chemical composition, size distributions, and the impact of convection"

_Atmospheric Chemistry and Physics, 2020_

## Referee Comment (RC1) · Anonymous Referee #1 · 3 Nov 2020

This paper covers airmass origin and aerosol composition in support of the CAMP2Ex campaign in Southeast Asia. The methods are sound and the paper is very well written indeed. While the location for the measurements is an area that has not received much attention over the years, I find this paper somewhat lacking in that it only presents results without any scientific implications to speak of. Specifically, I would regard none of the conclusions listed (a change in synoptic transport with monsoon onset, changes in emissions with geographic region and evidence of scavenging) to be particularly novel from a scientific perspective when presented on their own. As such, I do not

find this paper publishable in its current form; I recommend that it should either: 1) Be reclassified as a 'measurement report' or 2) Present new findings relevant to current atmospheric science in the discussion and conclusions.

Besides this fundamental issue, I could only find very minor points as follows:

Section 2.1: Please give information on the inlet system used for the aerosol instruments.

Line 121: What elevation data is the Google API using? USGS?

Line 136: Where does the uncertainty of 10% come from? Authors should also specify the calibrant used for the SP2, as this differs between groups.

Line 155: Presumably the NCEP reanalysis was used. If so, it should be specified as such.

Section 3.2: Not enough detail is given regarding the motivation and approach taken to perform the sensitivity analysis. This needs expanding on.

---

## Referee Comment (RC2) · Anonymous Referee #2 · 17 Dec 2020

The manuscript submitted by Hilario et al is a well written, technically sound analysis of the source regions and transport pathways impacting the airmasses sampled during the recent NASA CAMP2EX campaign. Statistical analysis of back trajectories from a meteorological reanalysis, together with precipitation satellite products are used to assign source regions and evaluate the possible impact of wet deposition/convection on these pathways. Several sensitivity analysis are performed to probe the robustness of the assignment, in particular in the vertical. Overall, with this approach the authors are able to assign a unique source region to 1/3 of airmasses sampled. The (fairly

limited) gas and aerosol instrumental package on the NASA P-3 is then used to support the regional assignments made, and to illustrate the effects of vertical motion and wet deposition upon these variables.

While the authors imply in the abstract and introduction that this work is highly relevant to the general interpretation of atmospheric chemistry measurements in the Equatorial Pacific, the synoptic results presented are really only relevant to the time of year of CAMP2EX (Sep/Oct), they are based on a heavily biased sample (by the flight plans of the research aircraft involved), and with only 33% of positive identification in that biased sample it seems hard to justify calling them representative for the region at large. The discussion of vertical transport and source mix is interesting and well written, but given the limitations of the payload they can at best confirm expected trends based on our current understanding. These results might set the stage to much more novel results gathered during CAMP2EX, but those results are clearly not part of the paper as written.

So to summarize, this is a very good description of the synoptic conditions for a particular subset of CAMP2EX measurements. Hence, I do agree with the other reviewer that, as written this would make a fine ACP measurement report and I would strongly support resubmission as such, once my detailed comments are addressed. Alternatively, the authors could shorten the manuscript significantly (lots of technical details could easily go into the SI to keep it at a reasonable length) and add some of the science that they propose in their next steps, and resubmit this as a research article.

Major comments: - This manuscript uses as lot of acronyms, many of which are fairly specific to this study/meterological context, which makes it at times fairly hard to follow. So I would suggest to include a simple lookup table, especially for the benefit of casual readers. - The whole discussion on the species ratios is very qualitative. While the authors emphasize a few times that they have very little information on the sources and sinks along the trajectories, the same is true for the emissions themselves, as currently written. The discussion would be considerably stronger if the authors had compiled

average emission ratios for each source region based on regional/global emission inventories (e.g. REAS2, CEDS), and comparing those numbers to the ratios actually observed. That would not only put the analysis on a more quantitative basis, but also make clearer how much contrast to actually expect for different air masses. In that context, I would encourage the authors to use one consistent denominator for emission ratios. E.g. while indeed peat burning has higher sulfate emissions than other fuels (Line 410), relative to fuel burnt (or CO), looking at the SO4/OA ratio is not really helpful, since we expect dOA/dCO to be higher in this case and dSO4/dCO lower. Using a consistent metric (also in Fig 7) would streamline the discussion and make quick comparisons easier. - On a similar note, the discussion of the size distributions (Fig 8) is fairly speculative and confusing (at least to me). The difference between clean FT and BL is clear cut and obvious, and does confirm the analysis of the vertical distribution by species/species ratio. But for the BL data, the current discussion focuses too much on possible differences in the source region and too little on processing, which especially when discussing the number distribution is a major concern. E.g. size distributions in large urban areas (so primarily EA) do typically have a large Aitken mode composed of fresh HOA/nitrate (e.g. see Zhang et al, 2005), but that smaller mode is short lived due to coagulation/aging and typically is not observed downwind of the urban core, let alone 96-120 h downwind, so I would be careful interpreting the MC/EA differences in this way (e.g most of this could be secondary sulfate from SO2 oxidation downwind, but again, without supporting evidence, this is just a guess). Similarly, the dip between Aitken and accumulation mode in the MBL has long been attributed to cloud processing (Hoppel et al, 1985), so trying to interpret the depth of that dip based on the source region seems like a stretch. Lastly, given the sampling region, the reported AMS + BC speciation may or more likely may not fully represent the total aerosol volume, hence complicating the interpretation further. So I would significantly shorten that part of the discussion, which without considerable additional information is just not well supported. One possible way to improve on the current discussion would be to map the AMS mass size distributions (should there be any and their S/N of high

enough quality) on the volume size distributions (Fig S9). E.g. if there are significant differences in composition at 80-120 nm between different regions, that would certainly be an interesting result and could give more insight into the processes involved. But given the low overall concentrations, this might be challenging. - Section 2.3/Figure 9: Using satellite data instead of e.g. the NCEP reconstructed meteorology along the back trajectories is a sensible choice, since the higher spatial resolution of most of these satellites products can be beneficial. However it is unclear as written how this four data products are merged and prioritized, especially given the very different time resolution of the individual products. This does not matter for the bulk of the analysis (where as Table 1 shows, the results for all approaches are compiled and give consistent results, with scatter roughly proportional to the fineness of the temporal resolution, as expected). But it is unclear which approach or combination of approaches went into the X-Axis of Figure 9a for the individual points shown, please elaborate. Related, it would be actually very useful to compare the different APT factors with the calculated convective and precipitation probability in the meteorological reconstruction used for the back trajectories. The performance of these reconstructions has steadily improved in recent years, so a brief assessment on how different the results would be if only relying on the model would be a valuable addition. - The scavenging mechanism for MC air in the FT is an interesting theory (also pretty speculative, as pointed out by the authors). However, the discussion does not sufficiently acknowledge that it's particles that are scavenged, not AMS species. And as the size distribution data shows, there is not much evidence for external mixing of sulfate and OA for MC air in particular (again, AMS SD data would be helpful here). So scavenging is unlikely to be species specific. In addition, BBOA tends to age fairly fast, especially in a high OH environment such as the tropics. I would assume that AMS f60 is fairly low/almost gone by the time these airmasses were sampled. If so, that would suggest a fairly high AMS f44/ O/C ratio. Which in turn means that a) those particles are likely internally well mixed (see e.g. Gorkowski et al, 2020) and b) the OA is reasonable hygroscopic (e.g Brock et al, 2016). So again, this makes the fact that OA is not scavenged as well as sulfate fairly

puzzling and needs to be discussed in more detail. - I am not convinced that Fig 9b shows what the authors intend. My reading of Fig 9a is that to a large extent, the trend in dBC/dCO is driven by different emission ratios for each region (not all of it, which is why this is a useful figure). But showing the data in the aggregate as done in Fig 9b averages again over these differences and will be hence just reflect changes in source region, not wet deposition

Minor comments: - Abstract: As noted above, this study does advance our understanding of long range transport in the TWNP, but it does not show that most of this applicable outside the intensive CAMP2EX period. Hence I would disagree that this research can "...[guide] international policymaking on public health and climate" (also end of the Introduction) and would suggest rephrasing. - Intro, 2nd paragraph: Most of the chosen names for the source regions are not standard by any means, they are just good operational monikers chosen by the authors for this particular study. And in fact, they are introduced as such later (with coordinates), L105-L107. So I would suggest to the authors to just use country names at this stage, and refer to the operational definitions later (so e.g. "Korea and SE China, referred to as EA in the following). - Line 101: Given the environment and aerosol mix, I think the recent paper on convective scavenging by Yu et al (2019) is relevant as well: Yu, P., Froyd, K. D., Portmann, R. W., Toon, O. B., Freitas, S. R., Bardeen, C. G., Brock, C., Fan, T., Gao, R.-S., Katich, J. M., Kupc, A., Liu, S., Maloney, C., Murphy, D. M., Rosenlof, K. H., Schill, G., Schwarz, J. P. and Williamson, C.: Efficient In-Cloud Removal of Aerosols by Deep Convection, Geophys. Res. Lett., 46(2), 1061–1069, 2019. - Line 123: Please indicate the refractive index the optical calibration of the LAS is based on. For the uncertainty, please provide a reference or describe the uncertainty sources in more detail. Also, I would note that while the FIMS and SP2 have similar upper size limits, 1000 nm optical size (again, depending on the calibration used) is under most conditions more than the D50 for the AMS equipped with a PM1 lens, so at a minimum I would suggest adding the size ranges for each instrument and clarifying how 1000 nm Dopt compares to those. - Line 129: deCarlo et al (2008) is a field study (and neither the first AMS flight deployment). The best instrumental reference here is likely Canagaratna et al (2007) (for the AMS in general) or deCarlo et al (2006) (for the HR-AMS in particular, if this is indeed the instrument flown, please specify). Please include this information. - Line 133: It is unclear how the detection limits were estimated, are these from blanks or using the method from Drewnick et al (2009). Importantly, detection limits are only meaningful in combination with their sampling period. I assume this is for 30 s, based on the discussion, but please indicate. - Line 134: It is unclear what this statement is supposed to mean. Data under DL at e.g. 30 s sampling time does still contain information that can become meaningful when averaging for longer periods. If the data was rather zero'd, this would actually bias the data. Can the authors clarify? - Line 135: Again, please provide a cite/justify the uncertainty estimate. - Line 137: I would suggest adding an entry to Table 1 with the AMS totals excluding RF9 - Line 142: While this is a sensible choice for the optical instruments, the tropical pacific is very cloudy. Can the authors provide numbers on how this affected the data coverage? And how exactly was "cloudy" defined? - Line 159: While discussed later, convection along the trajectories should be mentioned here as well - Line 188: Isn't case 1 equivalent to e.g. long range transport from farther away source regions (e.g India, West Asia). If so, I would suggest making that explicit (also in the context of evaluating the impact of LTR on the CAMP2EX domain). It would also be informative if you could provide (even rough) percentages for cases 1-4. - Line 321: It is unclear what "unimodal" is supposed to mean here - Line 345: In the absence of better tracers, this is probably the best approach. How much data was screened this way? - Line 353: The altitude profiles are fairly coarse. A higher resolution altitude profile of temperature and RH would be better to establish the most sensible BL height for the sampled data. - Line 398: Again, in the absence of source emission ratios this is not very convincing. The marine continent has many biogenic and anthropogenic sources of $CO_2$ and CO that will contribute to the total. It is possible that the fire emissions overwhelm all that signal, but the authors have not made that case yet, so it is at least as likely an explanation as the ones they mention. - Figure S9: It would be much better to plot this with varying

y-scale and the total volume mentioned in the legend, it is very hard to actually see the differences for the cleaner sectors - Author contributions: Please be more specific on the individual contribution of each author, per ACP policy.

References

Zhang, Q., Canagaratna, M. R., Jayne, J. T., Worsnop, D. R. and Jimenez, J. L.: Time- and size-resolved chemical composition of submicron particles in Pittsburgh: Implications for aerosol sources and processes, Journal of Geophysical Research-Atmospheres, 110(D7), 2005.

Hoppel, W. A., Fitzgerald, J. W. and Larson, R. E.: Aerosol size distributions in air masses advecting off the east coast of the United States, J. Geophys. Res., 90(D1), 2365, 1985.

Gorkowski, K., Donahue, N. M. and Sullivan, R. C.: Aerosol Optical Tweezers Constrain the Morphology Evolution of Liquid-Liquid Phase-Separated Atmospheric Particles, Chem, 6(1), 204–220, 2020.

Brock, C. A., Wagner, N. L., Anderson, B. E., Attwood, A. R., Beyersdorf, A., Campuzano-Jost, P., Carlton, A. G., Day, D. A., Diskin, G. S., Gordon, T. D., Jimenez, J. L., Lack, D. A., Liao, J., Markovic, M. Z., Middlebrook, A. M., Ng, N. L., Perring, A. E., Richardson, M. S., Schwarz, J. P., Washenfelder, R. A., Welti, A., Xu, L., Ziemba, L. D. and Murphy, D. M.: Aerosol optical properties in the southeastern United States in summer – Part 1: Hygroscopic growth, Atmos. Chem. Phys., 16(8), 4987–5007, 2016.

Yu, P., Froyd, K. D., Portmann, R. W., Toon, O. B., Freitas, S. R., Bardeen, C. G., Brock, C., Fan, T., Gao, R.-S., Katich, J. M., Kupc, A., Liu, S., Maloney, C., Murphy, D. M., Rosenlof, K. H., Schill, G., Schwarz, J. P. and Williamson, C.: Efficient In-Cloud Removal of Aerosols by Deep Convection, Geophys. Res. Lett., 46(2), 1061–1069, 2019.
Canagaratna, M. R., Jayne, J. T., Jimenez, J. L., Allan, J. D., Alfarra, M. R., Zhang, Q., Onasch, T. B., Drewnick, F., Coe, H., Middlebrook, A., Delia, A., Williams, L. R., Trimborn, A. M., Northway, M. J., Decarlo, P. F., Kolb, C. E., Davidovits, P. and Worsnop, D. R.: Chemical and microphysical characterization of ambient aerosols with the Aerodyne Aerosol Mass Spectrometer, Mass Spectrom. Rev., 26(2), 185–222, 2007.

DeCarlo, P. F., Kimmel, J. R., Trimborn, A., Northway, M. J., Jayne, J. T., Aiken, A. C., Gonin, M., Fuhrer, K., Horvath, T., Docherty, K. S., Worsnop, D. R. and Jimenez, J. L.: Field-deployable, high-resolution, time-of-flight aerosol mass spectrometer, Anal. Chem., 78(24), 8281–8289, 2006.

Drewnick, F., Hings, S. S., Alfarra, M. R., Prevot, A. S. H. and Borrmann, S.: Aerosol quantification with the Aerodyne Aerosol Mass Spectrometer: detection limits and ionizer background effects, Atmospheric Measurement Techniques, 2(1), 33–46, 2009

---

## Author Comment (AC1) · 21 Jan 2021

Response: We thank the two reviewers for their thoughtful suggestions and constructive criticism that have helped us improve our manuscript. Below we provide responses to reviewer concerns and suggestions. A major change to note at the outset is that we are re-submitting the paper as a Measurement Report to alleviate concerns about the lack of novelty and/or scientific implications.

RC1:

[Figure]

Major Comments:

This paper covers airmass origin and aerosol composition in support of the CAMP2Ex campaign in Southeast Asia. The methods are sound and the paper is very well written indeed. While the location for the measurements is an area that has not received much attention over the years, I find this paper somewhat lacking in that it only presents results without any scientific implications to speak of. Specifically, I would regard none of the conclusions listed (a change in synoptic transport with monsoon onset, changes in emissions with geographic region and evidence of scavenging) to be particularly novel from a scientific perspective when presented on their own. As such, I do not find this paper publishable in its current form; I recommend that it should either: 1) be reclassified as a 'measurement report' or 2) Present new findings relevant to current atmospheric science in the discussion and conclusions.

Response: We emphasize that the current paper sets the stage for future papers on topics such as new particle formation, transport-related secondary formation, and 3-dimensional scavenging effects. This paper serves as a valuable reference for transport patterns across the equatorial Pacific during a highly dynamic time of year (i.e., sampling both southwesterly biomass burning and northerly urban emissions), highlighting the potential variability of air mass origin in this region. Given the limited aircraft-based studies on transport patterns in this region, we maintain that this paper provides a much-needed characterization of this highly complex and climate-sensitive region. To satisfy this reviewer's concern, we are submitting this paper as a "Measurement Report".

Minor Comments:

Besides this fundamental issue, I could only find very minor points as follows. Section 2.1: Please give information on the inlet system used for the aerosol instruments.

Response: We have provided the requested information in Section 2.1. The added text reads:

"All in-situ aerosol measurements were placed downstream of a forward-facing shrouded solid diffuser inlet designed by the University of Hawaii that efficiently transmits particles ($\leq 5.0\ \mu$m aerodynamic diameter) to cabin-mounted instrumentation (McNaughton et al., 2007). The inlet flow rate is manually controlled to provide isokinetic sampling over the full range of P-3B airspeeds to minimize size-dependent biasing of the ambient particle size distribution. Downstream of the inlet, flow is split to individual instruments using a custom-designed stainless-steel manifold (Brechtel Manufacturing Inc.)."

Line 121: What elevation data is the Google API using? USGS?

Response: Over the US, the USGS National Elevation Dataset (NED) is the dominant source of Google's API, with some additional higher resolution lidar data when available. However, over the western Pacific, Google is not entirely transparent about the exact source of its elevation data, which seems to be proprietary. We note that as the Google elevation data is freely available, direct comparisons with other studies are quite possible. Furthermore, any uncertainty in elevation is mitigated by the fact that a large percentage of vertical profiles in this study were done over water, thus any errors in the elevation would minimally affect results. As a result, we felt the best course was to not revise any existing text about this issue in the draft.

Line 136: Where does the uncertainty of 10% come from? Authors should also specify the calibrant used for the SP2, as this differs between groups.

Response: We have added a description of the uncertainty quantification and SP2 calibration in Section 2.1. The added text reads:

"Black carbon (BC; ng m-3) was measured with a Single-Particle Soot Photometer (SP2) (Moteki & Kondo, 2007, 2010), including an uncertainty of 15% based on laboratory intercomparison results from Slowik et al. (2007). Lower detection limits are less than 10 ng m-3 based on manufacturer specifications and confirmed by in-flight filter-blank measurements and observations in the clean tropical free troposphere.

SP2 mass concentration calibration is accomplished using monodisperse nebulized fullerene soot aerosol according to Gysel et al. (2011)."

Line 155: Presumably the NCEP reanalysis was used. If so, it should be specified as such.

Response: We added "reanalysis" to make this clearer.

Section 3.2: Not enough detail is given regarding the motivation and approach taken to perform the sensitivity analysis. This needs expanding on.

Response: We have expanded on the motivation and approach and described an example in greater detail for added clarification:

"Due to the complex nature of long-range transport and the limited resolution of the meteorological input data, there is inherent uncertainty in the generated trajectories. In order to assess the effect of this uncertainty on our results, we evaluated the effect of modifying the following variables on our source-region distribution: (1) trajectory height threshold over source regions; (2) back trajectory run time; (3) vertical profile filtering; (4) monsoon phase; and (5) aircraft sampling location. For example, to test the sensitivity of our results to trajectory height threshold (i.e., 2 km over source regions), we varied this threshold (e.g., 0.5, 1, 3 km over source regions), reclassified trajectories according to the new threshold, and compared the new source-region distribution to the original result, which was presented in Section 3.1."

---

## Author Comment (AC2) · 21 Jan 2021

Response: We thank the two reviewers for their thoughtful suggestions and constructive criticism that have helped us improve our manuscript. Below we provide responses to reviewer concerns and suggestions. A major change to note at the outset is that we are re-submitting the paper as a Measurement Report to alleviate concerns about the lack of novelty and/or scientific implications.

RC2:

[Figure]

-The manuscript submitted by Hilario et al is a well written, technically sound analysis of the source regions and transport pathways impacting the airmasses sampled during the recent NASA CAMP2EX campaign. Statistical analysis of back trajectories from a meteorological reanalysis, together with precipitation satellite products are used to assign source regions and evaluate the possible impact of wet deposition/convection on these pathways. Several sensitivity analysis are performed to probe the robustness of the assignment, in particular in the vertical. Overall, with this approach the authors are able to assign a unique source region to 1/3 of airmasses sampled. The (fairly limited) gas and aerosol instrumental package on the NASA P-3 is then used to support the regional assignments made, and to illustrate the effects of vertical motion and wet deposition upon these variables. While the authors imply in the abstract and introduction that this work is highly relevant to the general interpretation of atmospheric chemistry measurements in the Equatorial Pacific, the synoptic results presented are really only relevant to the time of year ofCAMP2EX (Sep/Oct), they are based on a heavily biased sample (by the flight plans of the research aircraft involved), and with only 33% of positive identification in that biased sample it seems hard to justify calling them representative for the region at large.

Response: We note that the 33% statistic excludes air masses passing through the Philippines and therefore that actual percentage of air impacted by transport is expected to be higher. We have edited the Abstract and Introduction to clarify that analyzed transport patterns refer to the southwest monsoon and transition periods and therefore implications refer to these specific times of the year. Furthermore, we have acknowledged that aircraft location inherently affects observed transport patterns in Sections 2.2 and 3.3.

(Abstract) "These results are important for understanding the transport and processing of air masses with further implications for modeling aerosol lifecycles and guiding international policymaking on public health and climate, particularly during the SWM and MT."

(Introduction) "By examining how transport and scavenging mechanisms impact air mass composition, our results have implications for improving the modeling of aerosol lifecycles during the SWM/MT in this meteorologically complex region. Furthermore, due to the health impacts of biomass burning and anthropogenic emissions, this work is also important for guiding policymaking related to public health and climate during the transport-intensive SWM/MT."

(Section 2.2) "We emphasize that these source region contributions represent frequencies of observation rather than frequencies of occurrence, as the sampling location of the aircraft introduces a bias inherent in aircraft campaigns."

(Section 3.3) "The relative contributions of source regions did vary significantly with sampling location, though areas surrounding Luzon (e.g., East of Luzon, North of Luzon) showed some degree of similarity. Thus, we emphasize that, as with any aircraft campaign, observed transport is to some degree dependent on aircraft location."

-The discussion of vertical transport and source mix is interesting and well written but given the limitations of the payload they can at best confirm expected trends based on our current understanding. These results might set the stage to much more novel results gathered during CAMP2EX, but those results are clearly not part of the papers written. So to summarize, this is a very good description of the synoptic conditions for a particular subset of CAMP2EX measurements. Hence, I do agree with the other reviewer that, as written this would make a fine ACP measurement report and I would strongly support resubmission as such, once my detailed comments are addressed. Alternatively, the authors could shorten the manuscript significantly (lots of technical details could easily go into the SI to keep it at a reasonable length) and add some of the science that they propose in their next steps, and resubmit this as a research article.

Response: As noted at the outset and in response to the other reviewer, we have changed the manuscript to be a "Measurement Report".

Major Comments:

-This manuscript uses as lot of acronyms, many of which are fairly specific to this study/meteorological context, which makes it at times fairly hard to follow. So I would suggest to include a simple lookup table, especially for the benefit of casual readers.

Response: We have added a lookup table (Table 1) and referenced it in the first paragraph of the Results and Discussion: "Due to the specificity of some acronyms used in this work, we have provided a lookup table with definitions (Table 1)."

- The whole discussion on the species ratios is very qualitative. While the authors emphasize a few times that they have very little information on the sources and sinks along the trajectories, the same is true for the emissions themselves, as currently written. The discussion would be considerably stronger if the authors had compiled average emission ratios for each source region based on regional/global emission inventories (e.g. REAS2, CEDS), and comparing those numbers to the ratios actually observed. That would not only put the analysis on a more quantitative basis, but also make clearer how much contrast to actually expect for different air masses.

Response: While an excellent idea, we feel this is best left for future work and is something that would complicate further if this paper is a Regular Article or a Measurement Report. Our decision to change to a Measurement Report intends to reserve more detailed analyses like the one suggested here for future endeavors.

-In that context, I would encourage the authors to use one consistent denominator for emission ratios. E.g. while indeed peat burning has higher sulfate emissions than other fuels(Line 410), relative to fuel burnt (or CO), looking at the $SO_4/OA$ ratio is not really helpful, since we expect $dOA/dCO$ to be higher in this case and $dSO_4/dCO$ lower. Using a consistent metric (also in Fig 7) would streamline the discussion and make quick comparisons easier.

Response: We have replotted Figure 7 to use consistent denominators for easier comparison between species ratios and updated the text accordingly.

- On a similar note, the discussion of the size distributions (Fig 8) is fairly speculative and confusing (at least to me). The difference between clean FT and BL is clear cut and obvious, and does confirm the analysis of the vertical distribution by species/species ratio. But for the BL data, the current discussion focuses too much on possible differences in the source region and too little on processing, which especially when discussing the number distribution is a major concern. E.g. size distributions in large urban areas (so primarily EA) do typically have a large Aitken mode composed of fresh HOA/nitrate (e.g. see Zhang et al, 2005), but that smaller models short lived due to coagulation/aging and typically is not observed downwind of the urban core, let alone 96-120 h downwind, so I would be careful interpreting the MC/EA differences in this way (e.g most of this could be secondary sulfate from SO2 oxidation downwind, but again, without supporting evidence, this is just a guess). Similarly, the dip between Aitken and accumulation mode in the MBL has long been attributed to cloud processing (Hoppel et al, 1985), so trying to interpret the depth of that dip based on the source region seems like a stretch. Lastly, given the sampling region, the reported AMS + BC speciation may or more likely may not fully represent the total aerosol volume, hence complicating the interpretation further. So I would significantly shorten that part of the discussion, which without considerable additional information is just not well supported. One possible way to improve on the current discussion would be to map the AMS mass size distributions (should there be any and their S/N of high enough quality) on the volume size distributions (Fig S9). E.g. if there are significant differences in composition at 80-120 nm between different regions, that would certainly be an interesting result and could give more insight into the processes involved. But given the low overall concentrations, this might be challenging.

Response: From the outset, we agree that without a deeper investigation, the interpretation of the size distribution data is somewhat speculative. Though AMS size distributions would indeed have enhanced the discussion (a great idea), these data were

not collected during the campaign (bulk only). In response to your comment, we have shortened the size distribution section, stressed the importance of processes versus source regions, and highlighted findings that may lead to future work. Specifically, we highlighted the role of aging processes during transport from EA, which explains the notably wide accumulation mode peak and the absence of an Aitken mode peak (Fig. 8a). We have also noted that similar growth may also occur in MC air masses, leading to the observed accumulation mode peak (Fig. 8b). In addition to shortening and clarifying the section, major changes to the section include:

1. Revised EA paragraph: "EA air in the BL (Fig. 8a) had a wide peak (40 − 200 nm), suggestive of contributions from multiple sources (e.g., industrial, rice cultivation) (Chen et al., 2020b; Geng et al., 2019; Wang et al., 2016; Xia et al., 2020). The width of the accumulation mode peak and the absence of an Aitken mode peak may indicate aged aerosol that have been shifted towards larger modes during transport (Zhang et al., 2005)."

2. Revised MC paragraph: "MC air in the BL (Fig. 8b) had a single peak centered at 100 nm. Biomass burning emissions have been shown to greatly influence air mass composition (Engling et al., 2014; Fujii et al., 2015; Santoso et al., 2011) and, by extension, such a dominant emission source in addition to growth processes during transport explain the large accumulation mode peak (Figs. 8b and S8b). The Aitken mode peak in MC FT air supports the possibility of new particle formation (NPF) and growth (Williamson et al., 2019), promoted by the removal of aerosols and transport of gases during lofting into the FT (Fig. S6). Significant differences between the size distributions of FT and BL air from the MC (Fig. 8b) point to a potential scavenging mechanism acting on MC air lofted into the FT (Section 3.3.1)."

- Section 2.3/Figure9: Using satellite data instead of e.g. the NCEP reconstructed meteorology along the back trajectories is a sensible choice, since the higher spatial resolution of most of these satellites products can be beneficial. However it is unclear as written how this four data products are merged and prioritized, especially given the

very different time resolution of the individual products. This does not matter for the bulk of the analysis (where as Table 1 shows, the results for all approaches are compiled and give consistent results, with scatter roughly proportional to the fineness of the temporal resolution, as expected). But it is unclear which approach or combination of approaches went into the X-Axis of Figure 9a for the individual points shown, please elaborate.

Response: Under Section 3.5, we have stated which precipitation product was used for Fig. 9:

"For simplicity, APT in Fig. 9 is derived solely from PERSIANN-CDR, as Table 2 shows no significant qualitative difference between SPPs."

-Related, it would be actually very useful to compare the different APT factors with the calculated convective and precipitation probability in the meteorological reconstruction used for the back trajectories. The performance of these reconstructions has steadily improved in recent years, so a brief assessment on how different the results would be if only relying on the model would be a valuable addition.

Response: Regarding the addition of APT factors derived from NCEP reanalysis, we believe that the usage of three well-established satellite precipitation products (IMERG, TMPA, PERSIANN-CDR) achieves our objective of capturing variability in APT between different datasets. Indeed, a comparison of satellite and NCEP precipitation would be useful; however, given the higher spatial resolution of satellite products compared to NCEP reanalysis (as pointed out by the reviewer), we believe such an analysis is outside the study's scope and could be conducted as part of a broader paper evaluating these products.

- The scavenging mechanism for MC air in the FT is an interesting theory (also pretty speculative, as pointed out by the authors). However, the discussion does not sufficiently acknowledge that it's particles that are scavenged, not AMS species.

[Figure]

Response: We have acknowledged that it is particles that are scavenged by noting BL-FT differences in terms of particle concentrations:

"Comparisons of BL and FT aerosol concentrations (Fig. S6) and size distributions (Section 3.4; Fig. 8b) further support this possibility, as aerosol and particle concentrations have significantly lower values in the FT compared to BL, a difference not observed for trace gases (Fig. S6)."

In the next comment, we explicitly state the assumption that these particles are externally mixed and also clarify that discrepancies between sulfate and OA in aged air masses require further study.

-And as the size distribution data shows, there is not much evidence for external mixing of sulfate and OA for MC air in particular (again, AMS SD data would be helpful here). So scavenging is unlikely to be species specific. In addition, BBOA tends to age fairly fast, especially in a high OH environment such as the tropics. I would assume that AMS f60 is fairly low/almost gone by the time these airmasses were sampled. If so, that would suggest a fairly high AMS f44/ O/C ratio. Which in turn means that a) those particles are likely internally well mixed (see e.g. Gorkowski et al, 2020) and b) the OA is reasonable hygroscopic (e.g Brock et al,2016). So again, this makes the fact that OA is not scavenged as well as sulfate fairly puzzling and needs to be discussed in more detail.

Response: As AMS SD data was not collected during the campaign, we have reduced the speculative strength of the hypothesized scavenging and elaborated further on the discrepancy between scavenged $SO_4^{2-}$ and non-scavenged OA. We note that a separate analysis of m/z ratios for transported air from the Maritime Continent is underway by other CAMP2Ex groups.

"The discrepancy between $\Delta SO_4^{2-}/\Delta CO$ and $\Delta OA/\Delta CO$ (Fig. S7) implies externally mixed particles, which is surprising given the aged nature of these air masses (Gorkowski et al., 2020). Further analysis is required involving m/z, O:C ratios to account for aging effects (e.g., internal mixing, oxidation) and determine the exact mechanism behind the difference. We emphasize that the hypothesized scavenging of MC air in the FT is largely speculation for now and mainly introduces opportunities for future work."

- I am not convinced that Fig 9bshows what the authors intend. My reading of Fig 9a is that to a large extent, the trend in dBC/dCO is driven by different emission ratios for each region (not all of it, which is why this is a useful figure). But showing the data in the aggregate as done in Fig 9baverages again over these differences and will be hence just reflect changes in source region, not wet deposition

Response: Emission ratios are expected to contribute to differences in dBC/dCO between source regions. We have edited the text to make this contribution explicit:

"The decrease in $\Delta$BC/$\Delta$CO ratio with higher APT (Fig. 9) indicates that convection during transport indeed contributes to scavenging in the TWNP, though we note that source emission ratios also play an important role in the $\Delta$BC/$\Delta$CO ratio"

We have also improved Fig. 9a by plotting curves per source region to show source-resolved trends in dBC/dCO as a function of APT. Given the improvement in Fig. 9a, we have opted to remove Fig. 9b as it does not contribute significantly to the discussion. The text has been edited accordingly (Section 3.5).

Minor comments: - Abstract: As noted above, this study does advance our understanding of long range transport in the TWNP, but it does not show that most of this applicable outside the intensive CAMP2EX period. Hence I would disagree that this research can "...[guide] international policymaking on public health and climate" (also end of the Introduction) and would suggest rephrasing.

Response: We have rephrased the Abstract and Introduction to better reflect the focus on the southwest monsoon and monsoon transition periods.

(Abstract) "These results are important for understanding the transport and processing

of air masses with further implications for modeling aerosol lifecycles and guiding international policymaking on public health and climate, particularly during the SWM and MT."

(Introduction) "By examining how transport and scavenging mechanisms impact air mass composition, our results have implications for improving the modeling of aerosol lifecycles during the SWM/MT in this meteorologically complex region. Furthermore, due to the health impacts of biomass burning and anthropogenic emissions, this work is also important for guiding policymaking related to public health and climate during the transport-intensive SWM/MT."

- Intro, 2nd paragraph: Most of the chosen names for the source regions are not standard by any means, they are just good operational monikers chosen by the authors for this particular study. And in fact, they are introduced as such later (with coordinates), L105-L107. So I would suggest to the authors to just use country names at this stage, and refer to the operational definitions later (so e.g. "Korea and SE China, referred to as EA in the following).

Response: We have changed the monikers to country names.

- Line101: Given the environment and aerosol mix, I think the recent paper on convective scavenging by Yu et al (2019) is relevant as well: Yu, P., Froyd, K. D., Portmann, R. W.,Toon, O. B., Freitas, S. R., Bardeen, C. G., Brock, C., Fan, T., Gao, R.-S., Katich, J.M., Kupc, A., Liu, S., Maloney, C., Murphy, D. M., Rosenlof, K. H., Schill, G., Schwarz,J. P. and Williamson, C.: Efficient In-Cloud Removal of Aerosols by Deep Convection,Geophys. Res. Lett., 46(2), 1061–1069, 2019.

Response: We have included added it as a citation in the commented sentence.

- Line 123: Please indicate the refractive index the optical calibration of the LAS is based on. For the uncertainty, please provide a reference or describe the uncertainty sources in more detail. Also, I would note that while the FIMS and SP2 have similar

upper size limits, 1000 nm optical size(again, depending on the calibration used) is under most conditions more than the D50for the AMS equipped with a PM1 lens, so at a minimum I would suggest adding the size ranges for each instrument and clarifying how 1000 nm Dopt compares to those.

Response: We have added the following sentences to Section 2.1:

" Uncertainty of LAS-derived N100-1000nm is estimated at 20%. LAS optical sizing is calibrated using polystyrene latex spheres (i.e. with a real refractive index of 1.59) and verified in-flight using an onboard nebulizer to ensure consistent response throughout the campaign. During post-flight processing, sizing is corrected using monodisperse ammonium sulfate aerosol so that derived size distributions are referenced to a real refractive index of 1.53 and relevant to ambient aerosols (Shingler et al., 2016)."

"Particle size ranges for SP2 and AMS are reported at 100 – 700 nm (BC-equivalent), and 60 – 600 nm (vacuum aerodynamic) diameter, respectively. While quantitative comparison of these instruments is complicated by differing sizing techniques, each is relevant to accumulation-mode aerosol and are assumed to capture the majority of particle mass in this size range. Likewise, LAS integrated number concentrations from 100 – 1000nm optical diameter are used to illustrate variability in accumulation-mode number concentration."

- Line 129: deCarlo et al (2008) is a field study (and neither the first AMS flight deployment). The best instrumental reference here is likely Canagaratna et al (2007) (for the AMS in general) or deCarlo et al (2006) (for the HR-AMS in particular, if this is indeed the instrument flown, please specify). Please include this information.

Response: We have cited them in the commented sentence.

- Line133: It is unclear how the detection limits were estimated, are these from blanks or using the method from Drewnick et al (2009). Importantly, detection limits are only meaningful in combination with their sampling period. I assume this is for 30 s, based

on the discussion, but please indicate.

Response: We have rephrased the text for clarity. The updated text now reads:

"The AMS was operated in 1 Hz Fast-MS mode and averaged to 30-second time resolution for this study. 1-sigma detection limits (in $\mu$g m-3) are as follows for the 30-second averaged data: 0.169 (OA), 0.039 (SO42-), 0.035 (NO3-), 0.169 (NH4+). Detection limits were determined in-flight when sampling behind a filter-blank or during periods in the free troposphere with constant aerosol concentrations."

- Line 134: It is unclear what this statement is supposed to mean. Data under DL at e.g. 30 s sampling time does still contain information that can become meaningful when averaging for longer periods. If the data was rather zero'd, this would actually bias the data. Can the authors clarify?

Response: To clarify, we did include the negative AMS values in our analysis to avoid the positive bias the reviewer referred to. We have added a sentence in the text to make this clear:

"Mass concentrations below these detection limit values, which are sometimes negative due to the AMS difference method, are statistically equal to zero. To avoid a positive bias, negative AMS values were included in the analysis but were interpreted as concentration value of zero (e.g., Table 2)."

- Line135: Again, please provide a cite/justify the uncertainty estimate.

Response: The species of relevance to this study include sulfate (SO42-), ammonium (NH4+), nitrate (NO3-), and organic aerosol (OA), all of which are reported in units of $\mu$g m-3. AMS mass concentrations are reported with an uncertainty of up to 50% to account for ambiguity in the instrument collection efficiency. We have added this reasoning to the text.

- Line 137: I would suggest adding an entry to Table 2 with the AMS totals excluding RF9

Response: We have added the entry to the table (now Table 3) and rephrased the commented sentence to reflect the change:

"We note that BC data were unavailable during one flight covering a major Borneo smoke event (RF9); thus, we have provided statistics of the AMS total minus RF9 for a more direct comparison with BC (Table 3)."

- Line 142: While this is a sensible choice for the optical instruments, the tropical pacific is very cloudy. Can the authors provide numbers on how this affected the data coverage? And how exactly was "cloudy" defined?

Response: We have updated the text to include these details:

"Only data collected via isokinetic sampling (McNaughton et al., 2007) were used to eliminate sampling artifacts from the shattering of large water and ice particles (Murphy et al., 2004). When the aircraft entered clouds, sampling was manually switched to a Counterflow Virtual Impactor (CVI) inlet (Brechtel Manufacturing Inc.). Using only data collected during isokinetic sampling removed 16% of CAMP2Ex samples."

- Line 159: While discussed later, convection along the trajectories should be mentioned here as well

Response: We have included convection along the trajectory in the sentence:

"We note here that "source region" refers to the attributed origin of an air mass as identified by our trajectory classification scheme and does not preclude the possibility of entrainment from emission sources during transport (e.g., shipping) nor small-scale convection along trajectories unresolved by GFS".

- Line 188: Isn't case 1 equivalent to e.g. long range transport from farther away source regions (e.g India, West Asia). If so, I would suggest making that explicit (also in the context of evaluating the impact of LTR on the CAMP2EX domain). It would also be informative if you could provide (even rough) percentages for cases 1-4.

Response: We have added a sentence after the commented one that reads:

"We note that the first scenario is equivalent to long-range transport from further away source regions not considered in this work (e.g., India, West Asia)."

We have also provided estimates of each component of the "Other" category: "As a consequence of our filtering scheme, a large portion of trajectories were tagged as "Other" (66.8%). This is attributable to several scenarios, including but not limited to: (1) air masses that passed over source regions, but above our BL height threshold of 2 km (∼20%); (2) air masses influenced by the Philippines (i.e., air masses that stayed over the Philippines at < 2 km for more than 6 hours) (∼8%); (3) transport permutations that occurred too infrequently to provide robust statistics (∼3%); and (4) stagnant air masses that did not reach any source region (≤ 35%)."

- Line 321: It is unclear what "unimodal" is supposed to mean here

Response: We have rephrased the sentence for clarity:

"Sampling of EA and WP air were largely constrained to the BL, though EA air was sampled almost entirely below the 900 hPa level while WP air was more evenly sampled".

- Line 345: In the absence of better tracers, this is probably the best approach. How much data was screened this way?

Response: 8% of the data were screened by the local filter. We have added this detail to the text:

"To reduce the effect of local emissions, we excluded trajectories classified as influenced by the Philippines (PH). This filter screened approximately 8% of the data."

- Line 353: The altitude profiles are fairly coarse. A higher resolution altitude profile of temperature and RH would be better to establish the most sensible BL height for the sampled data.

[Figure]

Response: We feel as though this type of effort warrants its own separate study, which is in fact underway by other groups. This is not a trivial task to tackle with the airborne data, which is why we want to recommend that we can point out the limitation of our method in Section 2.2 and leave the extensive analysis for future work by other groups:

"We use a 2 km threshold to differentiate between BL and FT air based on climatologically-derived BL heights in this region (Chien et al., 2019). This inherently comes with a degree of uncertainty; however, we believe a conservative value of 2 km is sufficient for an overview study of this kind. An effort to determine CAMP2Ex BL heights is ongoing and warrants its own separate study."

- Line398: Again, in the absence of source emission ratios this is not very convincing. The marine continent has many biogenic and anthropogenic sources of CO2 and CO that will contribute to the total. It is possible that the fire emissions overwhelm all that signal, but the authors have not made that case yet, so it is at least as likely an explanation as the ones they mention.

Response: We have made the contribution of other sources in the Maritime Continent more explicit by adding a sentence after the commented one:

"The contribution of other CO or CO2 sources within the MC besides biomass burning may also explain the low correlation."

- Figure S9: It would be much better to plot this with varying y-scale and the total volume mentioned in the legend, it is very hard to actually see the differences for the cleaner sectors

Response: We have replotted the figure accordingly with a varying y-scale to better show distributions for cleaner sectors.

- Author contributions: Please be more specific on the individual contribution of each author, per ACP policy.

Response: The authors contribution now reads:

"MRAH performed the analysis and prepared the manuscript. EC, MS, LZ, JPDG, GSD, EW, CER, JW, JZ, YW, SY, JF, SLA, AS for collection and preparation of the data. EC, MS, JSR, MOLC, JBBS, LZ, JPDG, GSD, PN, FJT, EW, JW, JZ, YW, AB, AS for input and feedback on the manuscript."
* * *